# Joint sequencing of human and pathogen genomes reveals the genetics of pneumococcal meningitis

John A. Lees ⓘ et al.#

*Streptococcus pneumoniae* is a common nasopharyngeal colonizer, but can also cause life-threatening invasive diseases such as empyema, bacteremia and meningitis. Genetic variation of host and pathogen is known to play a role in invasive pneumococcal disease, though to what extent is unknown. In a genome-wide association study of human and pathogen we show that human variation explains almost half of variation in susceptibility to pneumococcal meningitis and one-third of variation in severity, identifying variants in *CCDC33* associated with susceptibility. Pneumococcal genetic variation explains a large amount of invasive potential (70%), but has no effect on severity. Serotype alone is insufficient to explain invasiveness, suggesting other pneumococcal factors are involved in progression to invasive disease. We identify pneumococcal genes involved in invasiveness including *pspC* and *zmpD*, and perform a human-bacteria interaction analysis. These genes are potential candidates for the development of more broadly-acting pneumococcal vaccines.

---

Streptococcus pneumoniae, or the pneumococcus, is the leading cause of bacterial meningitis, which is an important cause of mortality and morbidity worldwide despite advances in vaccination and treatment[1,2]. The case fatality rate of pneumococcal meningitis is 17–20% and unfavourable outcome occurs in 38–50% of cases[3]. S. pneumoniae is also the leading cause of pneumonia and bacteremia. Invasive disease is preceded by nasopharyngeal colonization[3].

Knowledge of the contribution of genetic variability of humans and invading pathogens to pneumococcal meningitis susceptibility could guide development of new vaccines preventing the progression from asymptomatic carriage to invasive disease, whereas genetic variation associated with disease severity may guide new clinical intervention strategies during treatment[4]. However, the overall effect of human genetics on pneumococcal meningitis is unknown—whether it affects the disease at all, and if so, which specific regions of the genome cause the effect. Historically, genetic association studies on bacterial meningitis have been held back by only assessing candidate genes, small sample sizes or poorly defined phenotypes[5]. More recent host genome-wide association studies (hGWASs; glossary Table 1) have found associations of a protective variant in complement factor H (CFH) for children with meningococcal meningitis in Europe[6], between a long non-coding RNA and pneumococcal meningitis in children in Kenya[7] and between CA10 and self-reported bacterial meningitis[8].

In terms of pathogen variability, it is well known that the pneumococcal polysaccharide capsule, which determines its serotype, contributes to invasive propensity[9,10]. The pneumococcal genome also encodes a variety of proteins that directly interact with the host, mostly to enhance colonization and avoid the host immune response[11]. Mouse models have shown that some antigens such as pspC (cbpA) enhance virulence but are not essential in invasive isolates[12,13]. While it is known that these antigens have a role in colonization and disease, whether sequence variation at these loci has an effect on pathogenesis in human disease remains unclear. Previous small association studies have additionally suggested a role for platelet binding[14] and arginine synthesis[15] in pneumococcal meningitis, and analysis of within-host variation found that sequence variation of dlt and pde1 are associated with pneumococcal meningitis[16,17].

Pathogen GWASs (pGWASs) provide a way to identify pneumococcal sequence variation associated with meningitis, independent of genetic background and in an unbiased manner. While GWAS is more challenging in bacteria than in humans due to strong population structure and high levels of pan-genomic variation, recent methodological advances have helped overcome these issues[18–20].

Here, we use data and samples from Dutch adults in our prospective and nationwide MeninGene cohort[3]. We have collected a large number of samples of both human and pathogen DNA from culture-proven cases of pneumococcal meningitis, along with detailed clinical metadata (Supplementary Tables 1 and 2). Genotyping and whole-genome sequencing of this collection allows us to undertake a combined GWASs of host (hGWASs) and pathogen (pGWASs) in pneumococcal meningitis. We further include an analysis of interaction effects in a joint GWAS and replicate our findings in additional cohorts of invasive pneumococcal disease (IPD; Fig. 1). Our analyses define the role of genetic variation of host and pathogen in pneumococcal meningitis.

## Results

**Multiple bacterial loci determine pneumococcal invasiveness.** We first calculated the heritability of susceptibility and severity due to pneumococcal genetics using the MeninGene cohort. The heritability corresponds to the proportion of phenotypic variation explained by genetics, in this case single-nucleotide polymorphism (SNP) variation in the bacterial core genome. We used an additive binary phenotype model, which does not model any potential effects of genetic interactions (either epistatic or with the environment)[21]. Under this model, we found that additive pneumococcal genetics explained much of variation in invasive propensity ($h^2_{SNP} = 70\%$) but not meningitis disease severity ($h^2_{SNP} = 0\%$). We also calculated that 31% of the phenotypic variance could be explained by the top ten principal components, which explained 67% of the total genetic variance. This suggests that lineage effects, as in those pneumococcal strains which are more likely to be sampled from an invaded niche, only partially account for the estimated heritability. It may therefore be possible to find additional specific locus effects.

These $h^2_{SNP}$ estimates suggest that invasive propensity is highly dependent on pneumococcal genome content but that disease outcome is not determined by natural variation of pathogen genetics. The latter is not surprising as invasive disease is an evolutionary dead end for the pathogen, so adaptations affecting virulence over the short course of infection are unlikely to be selected for. This is contrary to smaller studies suggesting that bacterial genotype may predict disease severity[22] but consistent with a meta-analysis finding no effect of pneumococcal serotype on the risk ratio of death from meningitis[23]. In our population, levels of antimicrobial resistance are low and rarely led to treatment failure. In populations where resistance leads to treatment failure, finding $h^2 > 0$ for severity could be expected. The high heritability estimated here is consistent with the fact that some serotypes are rarely found in invasive disease[9], and invasive lineages are genetically divergent from the rest of the population. It should also be noted that all meningitis isolates were once carriage isolates, which by our sampling scheme are effectively overrepresented in an invaded niche. Some of the poorly invasive yet highly transmissible strains usually observed in carriage may also be able to cause invasive disease, potentially meaning heritability is slightly underestimated.

Pneumococcal serotype is the current focus of vaccination. Using the same framework as for core SNPs, we calculated the proportion of variation in invasiveness that can be attributed to

### Table 1 Glossary of terms

| Term | Meaning |
|---|---|
| CSF | Cerebrospinal fluid |
| IPD | Invasive pneumococcal disease |
| GWAS | Genome-wide association study |
| hGWAS | Host genome-wide association study |
| pGWAS | Pathogen genome-wide association study |
| $h^2$ | Heritability (variation in phenotype explained by genetic variation) |
| OR | Odds ratio |
| LD | Linkage disequilibrium |
| MAF | Minor allele frequency (of the least common allele observed in the population) |
| AF | Allele frequency (presence of gene or versus reference allele) |
| LoF | Loss of function (frameshift or nonsense mutation in protein) |
| SFS | Site frequency spectrum (histogram of minor allele frequencies) |
| LMM | Linear mixed model (association model used in genome-wide association study) |
| eQTL | Expression quantitative trait loci (genetic association with transcript variation) |
| LD | Linkage disequilibrium |

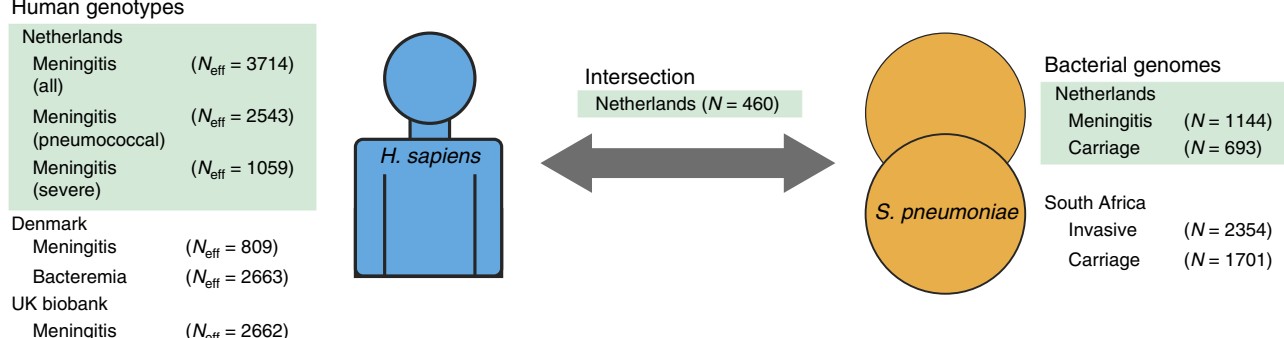

**Fig. 1** Overview of cohorts sequenced and associations performed. Left, host data; right, bacterial data; the centre represents samples with both host and pathogen sequence data. Samples in green are those collected from our MeninGene cohort that form the centre of this work. Owing to unbalanced case–control ratios, we show the effective sample size, specific numbers of cases and controls of human genotypes in Supplementary table 2

serotype collectively, finding $h^2_{serotype} = 50\%$ (77% of $h^2_{SNP}$). A model including both serotype and other genetic factors was supported over non-serotype relations alone (likelihood ratio test $p < 10^{-10}$), but decomposing these variance contributions quantitatively was not possible due to their correlation. Overall, this provides some evidence that serotype affects invasive potential independently of genetic background, but the strength of this conclusion is ultimately limited due to the covariance of these two factors. However, in considering whether serotype is the dominant factor in invasiveness, $h^2_{serotype} < h^2_{SNP}$ means we can more confidently expect that other pneumococcal loci are also involved in invasion.

**A potential role of rare variation in pneumococcal invasion**. We evaluated differences in the sequence variation between asymptomatic carriage and meningitis isolates. The amount of rare variation compared to common variation is informative of recent selection and population size changes[24]. An overall difference may be informative of different selection on regions of the genome depending on the niche. In Fig. 2a, we have plotted the site-frequency spectrum by niche and predicted consequence. Across the range of common minor allele frequencies (MAFs) in both niches, the proportion of synonymous/nonsynonymous/intergenic/loss-of-function (LoF) mutations is as previously observed[25]; at low frequencies, there is an excess of potentially damaging variants. Figure 2c shows the overall burden of damaging rare variants between carriage and invasive samples; in both LoF and damaging variants, there was higher burden in carriage isolates (median LoF: invasive 7, carriage 11, two-tailed Wilcoxon rank-sum test $W = 303{,}070$, $p < 1 \times 10^{-10}$; median damaging: invasive 22, carriage 26, Wilcoxon rank-sum test $W = 350{,}690$, $p = 3 \times 10^{-5}$). When correcting for the number of synonymous mutations, treated as neutral, in each sample there was still a higher excess burden of rare LoF and damaging variants in carriage isolates (damaging: two-tailed Wilcoxon rank sum test $W = 301{,}820$, $p < 1 \times 10^{-10}$; LoF: two-tailed Wilcoxon rank-sum test $W = 357{,}220$, $p = 0.001$). In the absence of population size change, this implies that certain genes are under stronger purifying selection in invasive lineages, with damaging mutations which still allow carriage being purged in the invasive population. We therefore attempted to quantify the amount of phenotypic variation due to the burden of rare damaging and LoF mutations but were unable to directly find significant evidence for susceptibility $h^2_{burden} > 0$ for any individual gene or over all genes. However, this calculation is underpowered due to the small number of rare mutations.

Instead, to quantify the difference in rare variants between invasive and carriage samples and to identify which regions of the

genome are responsible for the excess of rare alleles we calculated Tajima's $D$, a statistic for determining departures from neutral evolution, for each coding sequence in the genome, and looked for differing signs of selection between cases and controls. Deviations with $D < 0$ are indicative of selective sweeps and/or recent population expansion, whereas $D > 0$ is indicative of balancing selection and/or recent population contraction. We compared the distributions of $D$ by gene in each phenotype (Fig. 2b). Genes in invasive isolates had a lower average $D$ (difference in medians $-0.34$; two-tailed Wilcoxon rank-sum test $W = 1996100$, $p < 10^{-10}$) and a more positively skewed $D$ (difference in skewness 0.30; 95% bootstrapped confidence interval 0.17–0.44). This difference in $D$ may be representative of a genuine difference in selection of variants in genes between niche or may be due to a difference in recent population dynamics, for example, due to the bottlenecks for invasion and transmission. It is illustrative to evaluate these findings considering invasive population as a sample from the carriage population, with potentially invasive genotypes amplified. A greater average $D$ is not observed, as would be caused by rare variants becoming common through random subsampling through a bottleneck. The finding of a lower average $D$ would not necessarily always be expected from subsampling low diversity invasive clones from the overall population. Instead, the shift towards lower MAFs suggests that some allelic variants or genotypes found in carriage are disadvantageous in invasive disease or that recent, rare mutations may play a role in invasion.

**Identification of specific pneumococcal invasiveness loci**. We then tried to find pneumococcal factors other than serotype that are associated with disease. We first performed a pGWAS with disease severity in the Dutch MeninGene cohort. Consistent with our estimates of no heritability, we found no loci of any type to be significantly associated with severity.

We then analysed meningitis versus carriage isolates. We first analysed each of our cohorts individually (Supplementary Tables 4–6), and in an effort to make our results applicable across more of the species, we also combined the Dutch cohort of meningitis and carriage samples with a cohort collected in South Africa. This cohort included samples from carriage and cases of IPD. These isolates were from a heterogeneous population, around 15% of which were meningitis cases. While not an identical phenotype to the meningitis cases collected in the Dutch cohort, these phenotypes do overlap (all meningitis cases are IPD cases by definition). It was also recently shown, for a single population, that pneumococcal genomes from non-meningitis IPD are highly similar to meningitis IPD[26]. However, as there are many different infection routes, this approach will only be

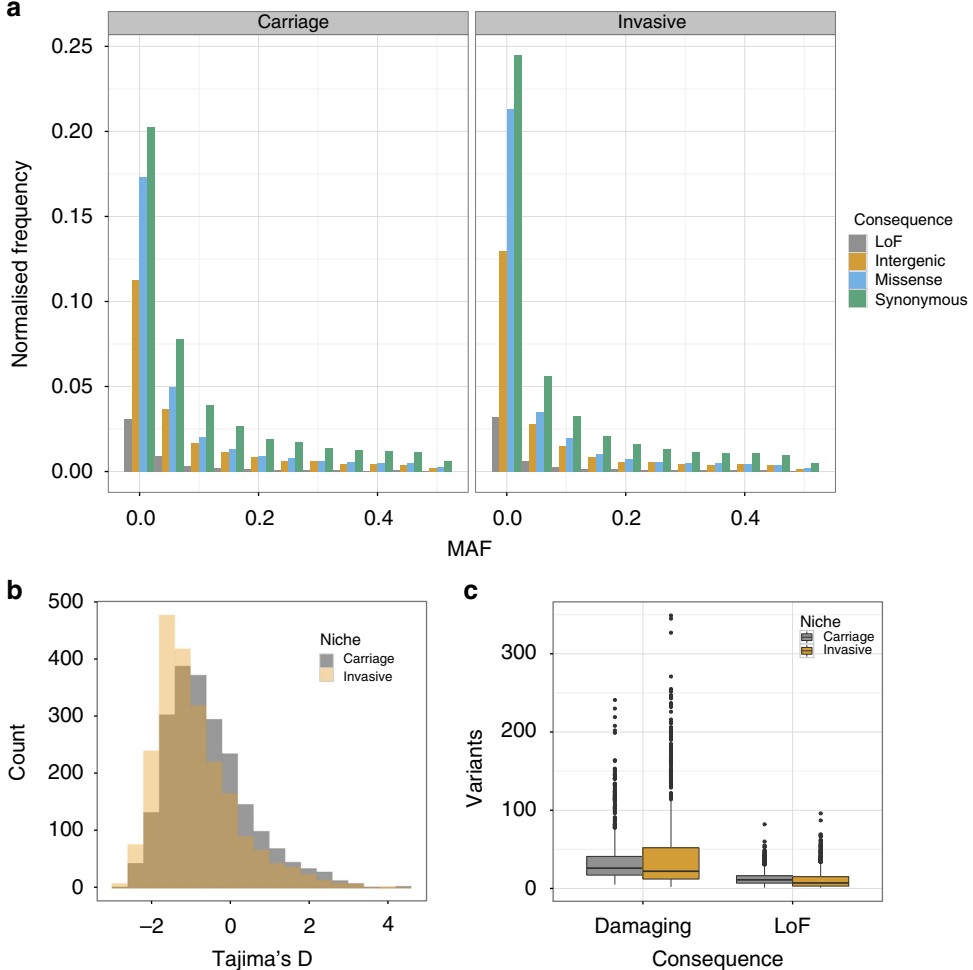

**Fig. 2** Burden of rare variation between invasive and carriage isolates, based on mapping and calling short variants against a single reference genome. Loss-of-function (LoF) are frameshift or nonsense mutations. **a** The site frequency spectrum (SFS) stratified by niche and by predicted consequence. Frequency has been normalized with respect to the number of samples in each population. **b** Histogram of Tajima's D for all coding sequences in the genome, stratified by niche. **c** Boxplot of the number of rare variants per sample, stratified by niche and predicted consequence. Damaging mutations are LoF mutations and missense mutations predicted damaging by PROVEAN. Centre line is the median, box spans lower to upper quartiles. Whiskers show the outlier range, defined as being >1.5× the interquartile range above or below the lower and upper quartiles

powered to detect those routes common between the cohorts—most likely invasion into the bloodstream.

This pooling of cohorts gave a total of 5845 pneumococcal genomes to analyse (Fig. 3). Carriage and invasive isolates (cases and controls) are distributed across the tree, though with clear clusters also associated with a single serotype. The two cohorts were genetically separated, with lineages generally at very different frequencies in each population. We controlled for this in two ways. First, by performing a pooled pGWAS both with the sample cohort as a covariate in our analysis and a co-estimated kinship matrix as random effects (Table 2). We also combined separate association studies in our two cohorts using a meta-analysis (Table 3), which more easily allows difference in direction and size of effect in each cohort to be seen.

Table 2 shows the genes that were significant in this combined analysis using any of our association methods – due to cohort heterogeneity they should be considered as hypotheses for association with natural IPD. The genes noted in bold font in Table 2 are all immunogenic[27] and have been associated with pneumococcal virulence in animal models[12,28–30] but not yet in patients with invasive disease. A k-mer in *pspC*, significant in the pooled analysis, was significant in the South African cohort but not in the Dutch cohort. However, we found that the presence of

other non-overlapping k-mers in *pspC* in each cohort was associated with invasive disease with the same direction of effect (maximum p-value $p_{SA} = 9.7 \times 10^{-13}$; $p_{NL} = 1.3 \times 10^{-9}$). We predicted that both forms of *pspC* were absent in 13 meningitis isolates, though manual inspection of the summary statistics from mapping and assembly suggested these may also be an unresolved form of allele 8. Previous conclusions, drawn from protein binding to the laminin and polymeric immunoglobulin receptors, have suggested that *pspC* (*cbpA*) is necessary for meningitis[31]. All 13 patients infected by these strains had a severe ear, nose or throat infection, suggestive for direct spread of bacteria rather than crossing the blood–brain barrier. Three patients had clear bone destruction and/or pneumocephalus and one patient had a skull defect due to previous surgery. We further tested whether the two major forms of PspC were associated with meningitis, as has previously been suggested[32], but did not find either to be overrepresented, when accounting for population structure. *dacB* is involved in preserving cell wall shape and has shown to attenuate virulence in a mouse model of lung infection[28]. This gene is highly expressed in early stages of infection, until expression declines after 2 hrs[33,34]. *zmpD* is homologous to IgA1 protease (*zmpA*)[35], and while it is immunogenic[27], its function is unknown. A previous study in asymptomatic carriage found it to

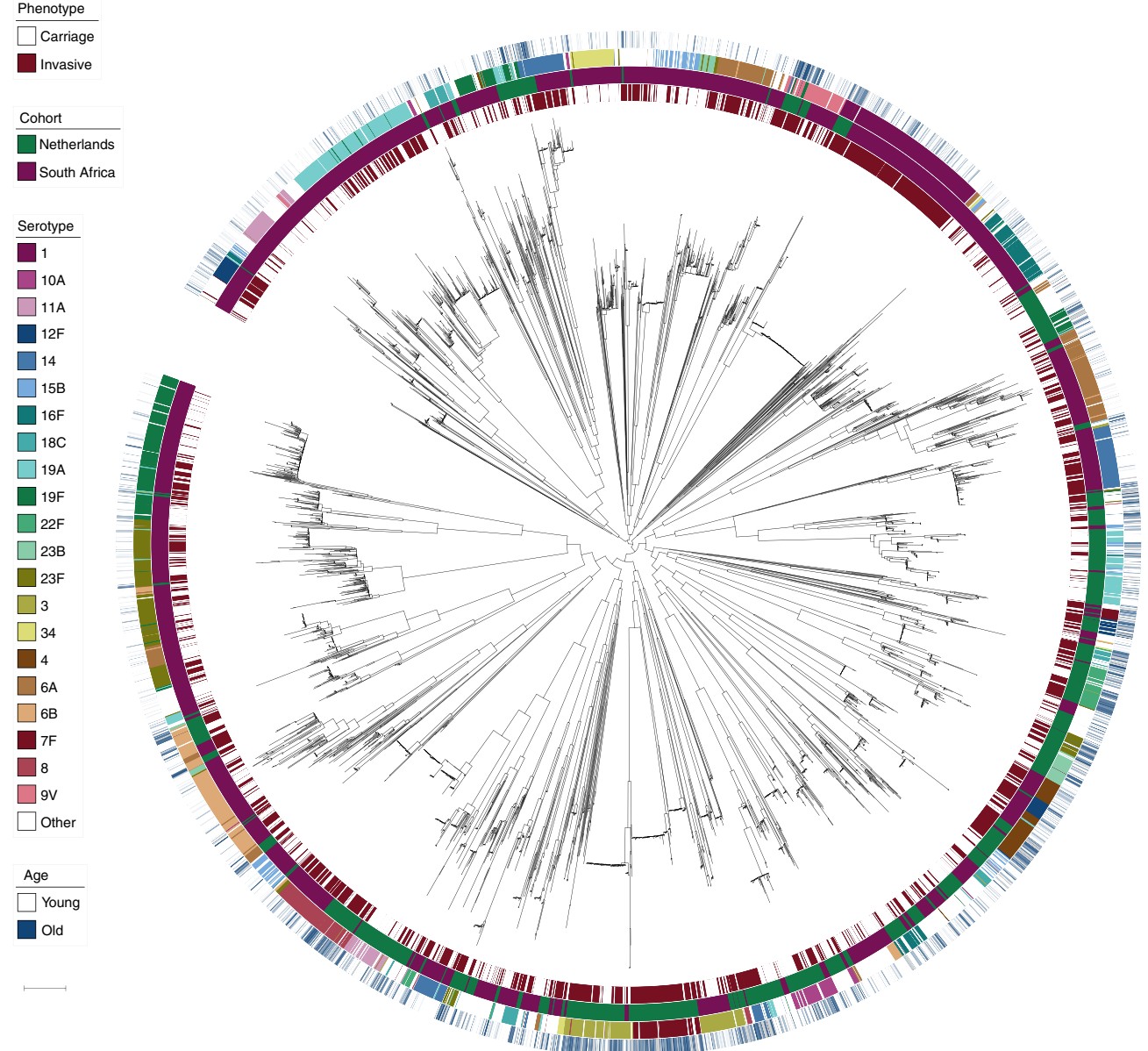

**Fig. 3** Phylogenetic tree of all samples included in the pathogen genome-wide association study. Rings show metadata about samples, from inside to outside: phenotype (carriage or invasive); cohort (Netherlands or South Africa); common serotypes; patient age on a continuous scale from younger (white) to older (blue). Scale bar: 0.01 substitutions per site. An interactive version is available at https://microreact.org/project/Spn_GWAS/9eb0bd5d (project link https://microreact.org/project/Spn_GWAS)

be more prevalent in older children[36]. The results from our pGWAS suggest a role in human cases of disease. This result seems to primarily be driven by singleton variants observed in invasive samples, which occur repeatedly at the tips of the tree (Supplementary Fig. 1).

The other genes in Table 2 have not previously been directly associated with virulence or invasive disease in *S. pneumoniae*. SPN23F09820 (FM211187.3090) is a protein of unknown function, which is annotated as bacteriocin precursor. There was no difference in its expression between simulated blood, cerebrospinal fluid (CSF) and infection conditions[33]. While different alleles of bacteriocins are known to be involved in cell–cell bacterial competition, whether a relation between allelic variation and pathogenesis exists is unknown. SPN23F05670 (FM211187.1804) is usually directly upstream of *dacB*, which it is coexpressed with, and therefore likely represents the same signal.

Other pneumococcal proteins containing the histidine triad (HIT) motif are expressed on the cell surface and bind human complement[37], but little is known about the specific function of this protein or its relationship with disease. When compared to the nasopharynx, *ecsA* is downregulated in CSF, blood and infection conditions. While in other species this gene has been associated with cell–cell interactions[38] and virulence[39], in *S. pneumoniae* allelic variants of *ecsA* have been previously correlated with serotype in carriage[40], so this may be linked with the effect of capsule. Finally, mutations in *mcrB*, an endonuclease that is more highly expressed in infection conditions, have been shown to lead to phage resistance[41]. Phages may have a number of different effects on the bacterial host cell, including loss of competence, changes in fitness or bringing in virulence cargo genes, but as we did not detect any of these elements directly through our pGWAS we cannot speculate further.

**Table 2 Signals of bacterial association in a pooled analysis**

| Gene ID | Gene name | Gene population frequency | Method | OR/effect size | p-value | Function |
|---|---|---|---|---|---|---|
| **SPN23F05680** | **ldcB/dacB** | **0.81** | **Missense burden in invasive** | **1.20** | **$1.3 \times 10^{-19}$** | **D-alanyl-D-alanine carboxypeptidase (peptoglycan peptide precursors)** |
| **SPN23F22240** | **pspC/cbpA** | **0.59** | **K-mers** | **1.05** | **$3.8 \times 10^{-13}$** | **Binds secretory IgA, C3 and complement factor H; adhesin** |
| **SPN23F08080** | **spnTVR hsdS** | **1.00** | **Missense burden in invasive** | **1.08** | **$2.8 \times 10^{-6}$** | **Type I restriction-modification system specificity subunit S adhesin** |
| **SPN23F17820** | **psrP** | **0.42** | **Tajima's D** | **−2.71 (invasive) −2.59 (carriage)** | **$<1 \times 10^{-6}$** | |
| SPN23F10590 | zmpD | 0.53 | Burden of rare LoF in invasive | 1.42 | $<1 \times 10^{-10}$ | Unknown; paralogous to IgA1 protease (zmpA) |
| SPN23F09820 | FM211187.3090 | 0.36 | Missense burden in invasive | 1.30 | $3.4 \times 10^{-49}$ | Bacteriocin precursor |
| SPN23F05670 | FM211187.1804 | 1.00 | Missense burden in invasive | 1.27 | $4.3 \times 10^{-47}$ | Histidine triad family protein (nucleotide phosphatase) |
| SPN23F04740 | ecsA | 1.00 | Missense burden in invasive | 1.15 | $1.4 \times 10^{-8}$ | ABC transporter ATPase |
| SPN23F11460 | mcrB | — | Missense burden in invasive | 1.13 | $1.2 \times 10^{-6}$ | Endonuclease |

Combining The Netherlands (meningitis only) and South African (all IPD cases) cohorts into a single dataset and performing a pGWAS with cohort as a covariate. Table shows genes significant (after applying a Bonferroni correction for the number of tests and association methods $p < 0.05$) in a pooled analysis of both cohorts with any of the association approaches, ordered by $p$-value. Odds ratios are with respect to carriage samples. The four genes in bold at top of the table are immunogenic and have previous evidence for association with virulence. For Tajima's $D$, the effect size is the difference between $D$ values, and for k-mers and LoF burden tests it is the odds ratio. For some $p$ values, the calculation only allows an upper bound to be produced. The locus tag in the ATCC 700669 reference is listed, along with the common gene name if available
OR odds ratio

**Table 3 Meta-analysis of signals of bacterial association**

| Gene ID | Gene name | OR (NL) | p-value (NL) | OR (SA) | p-value (SA) | Combined p-value |
|---|---|---|---|---|---|---|
| SPN23F05680 | ldcB/dacB | 1.39 | $2.8 \times 10^{-19}$ | 1.13 | $9.6 \times 10^{-8}$ | $5.4 \times 10^{-21}$ |
| SPN23F22240 | pspC/cbpA | 1.00 | $8.1 \times 10^{-1}$ | 1.12 | $4.7 \times 10^{-12}$ | $5.3 \times 10^{-11}$ |
| SPN23F08080 | spnTVR hsdS | 0.82 | $1.9 \times 10^{-4}$ | 1.09 | $4.2 \times 10^{-6}$ | $7.5 \times 10^{-2}$ |
| SPN23F17820 | psrP | −2.27 (invasive) −0.14 (carriage) | $<1 \times 10^{-6}$ | −2.61 (invasive) −2.62 (carriage) | $4.5 \times 10^{-4}$ | $<1 \times 10^{-6}$ |
| SPN23F10590 | zmpD | 1.17 | $8.6 \times 10^{-13}$ | 1.12 | $2.5 \times 10^{-5}$ | $8.4 \times 10^{-14}$ |
| SPN23F09820 | FM211187.3090 | 1.37 | $5.0 \times 10^{-23}$ | 1.30 | $1.9 \times 10^{-33}$ | $3.4 \times 10^{-54}$ |
| SPN23F05670 | FM211187.1804 | 1.47 | $1.8 \times 10^{-30}$ | 1.22 | $2.9 \times 10^{-25}$ | $8.3 \times 10^{-51}$ |
| SPN23F04740 | ecsA | 1.20 | $1.8 \times 10^{-7}$ | 1.11 | $9.7 \times 10^{-4}$ | $1.8 \times 10^{-8}$ |
| SPN23F11460 | mcrB | 1.45 | $3.7 \times 10^{-8}$ | 1.11 | $9.5 \times 10^{-5}$ | $1.2 \times 10^{-6}$ |

As in Table 2, showing behaviour of signals in The Netherlands (NL) and South African (SA) cohorts individually. The $p$ value is from meta-analysis of the two cohorts using METAL to compute a combined $z$ value, so is different from the $p$ value in Table 2, which was computed by applying an LMM to all samples
OR odds ratio

We also observed hits (not shown in Tables 2 and 3) to simple transposons without cargo genes, which we therefore discarded as we assumed this was an artefact of their independence from population structure. We also found significant association of some BOX repeats[42], but as there are many copies we could not map these associations to a single region of the genome.

It is notable that many of our pGWAS results were due to rare genetic variation. This variation may be more important in a disease like meningitis where there is no pervasive selection for the phenotype and effects are likely polygenic, as opposed to the strong monogenic effects seen for many antimicrobial resistance traits. Mutations with the same functional effect that occur at the tips of the tree are less confounded by population structure and therefore a signal that is easier to find with pGWAS. In the missense burden test, we are able to combine lineage variants that are heavily population stratified and therefore contribute only a small amount of information in relation to the number of samples they are found in, with functionally similar mutations elsewhere in the tree. Overall the invasive disease-associated genes had no propensity to accumulate missense variants (median dN/dS core genes = 0.25; median dN/dS hit genes = 0.28; Wilcoxon rank-sum test $W = 2741$, two-tailed $p$ value = 0.9414), suggesting the power increase is more likely due to population structure rather than mutation frequency. If these repeated functional changes were the only biological explanation for invasiveness, this would suggest that there exists a group of protein alleles that are more able to cause invasive disease, consistent with the Tajima's $D$ results above.

**Human genetics associates with susceptibility and severity**. In contrast to bacteria, human variation within populations of the same ancestry is not strongly confounded by population structure, and only sites in relatively close physical proximity show any correlation[43]. Mapping associations within the resolution of a single locus is therefore generally easier than in pGWAS, with LMMs again able to control for genetic background (in this case caused by differences in ancestry within the sampled population)[44]. Precise control of ancestry also allows for the comparison with genetic, transcriptomic and physical distance based associations from other cohorts, even when the ancestry of the cohorts are not exactly matched[45]. For these purposes, summary statistics (effect size, direction and $p$-value) of associations is often sufficient to combine data from different studies and data sources.

Being the most studied genome of any species, a wealth of information on human genes is available to further interpret genetic variants with relatively unknown function. Using data from the GTEx consortium (an expression quantitative trait locus (eQTL) study across 44 different tissues, wherein genetic variants are tested for association with changes in the expression levels across the transcriptome)[46,47], effects on expression on more distant genes or in specific tissues can be noted. Additionally, as eukaryotic DNA is tightly packed within the nucleus, longer-range interactions may exist between sites that are close in three-dimensional space but apparently distant along a one-dimensional genomic coordinate axis. These long-range interactions may, for example, function as enhancers, boosting the expression of distant genes in certain cell types. Chromatin conformation capture (Hi-C) data can find interaction partners across the genome in a range of cell types[48], which then may offer additional insight into effects of a variant beyond its immediate genomic region, and in which cells this interaction is important.

Using the human genotyping data we produced, we were also able to apply LMMs to calculate heritability due to host genetics and identify common SNP variation in humans that predisposes to the measured phenotypes. To acquire sufficient cases to accurately estimate heritability, we combined severe pneumococcal cases ($N_{cases} = 212$) with severe cases from any pathogen ($N_{cases} = 277$). To justify this, we reasoned that the enormous difference in host inflammatory response observed between these phenotypes may well be through the same host mechanism, irrespective of the specific infecting bacteria[3]. We then used two standard methods to calculate SNP heritability, which both showed that variation in host genetics explains 29% ± 7% of the observed variation in pneumococcal meningitis susceptibility and 49% ± 14% of the variation in meningitis severity (Table 4). These are relatively large genetic contributions compared to other infectious diseases[8].

Next, we performed a hGWAS to search for specific loci associated with meningitis. We did not find any associations

reaching significance for severe pneumococcal meningitis, but when combined with severe meningitis from all other species (as in the heritability calculations) one marker reached significance when testing for severity: position 64680775 (rs12081070) on chromosome 1, an intronic variant in *UBE2U* that was associated with unfavourable outcome (MAF = 0.43; odds ratio (OR) = 1.63; $p = 2.0 \times 10^{-8}$) (Supplementary Figs. 2–5). *UBE2U* is part of the ubiquitin pathway and is involved in antigen presentation through the class I major histocompatibility complex pathway[49] but has not been previously directly associated with any disease or trait in a genetic study.

We used existing data from chromatin conformation capture to find possible interaction partners of these loci in various cell types. This analysis showed that the site of the most significantly associated variant (rs12081070) interacts with *PGM1* and *ROR1* in a range of immune cell types, including monocyte/macrophages, CD4/8 T cells, B cells and neutrophils (Supplementary Fig. 5). *PGM1* encodes a phosphoglucomutase while *ROR1* is a protein of unknown function that has previously been associated with cancers[50] and pulmonary function[51]. Using the GTEx database of gene expression across different human tissues, we found evidence of association of rs12081070 with gene expression in a panel of tissues and cell types, but this was only significant in one (skin—$p = 5.7 \times 10^{-13}$)[47]. Six other loci showed suggestive significance (Table 5), whereas in the Danish cohort, no variants reached genome-wide significance (Supplementary Figs. 7 and 8).

Of the genes putatively implicated in meningitis in the MeninGene cohort at the suggestive level, we noted that the *ME2* promoter variant rs2850542 is an eQTL for the same gene in whole blood ($p = 5.9 \times 10^{-20}$, Supplementary Fig. 9A) and specifically in monocytes (false discovery rate $5.1 \times 10^{-26}$)[52]. There was also evidence of chromatin interaction of the variant location with *SMAD4* again in a range of immune cell types including monocytes, lymphocytes and neutrophils (Supplementary Fig. 9B, C). *SMAD4* is a key intracellular mediator of transcriptional responses to transforming growth factor beta and involved in the invasion across the epithelium; the variant also showed evidence of an eQTL involving rs2850542 for *SMAD4* expression in tibial artery ($p = 8.6 \times 10^{-7}$)[53]. These data tentatively suggest that the expression of these genes in the noted tissues, specifically blood, may be the means by which susceptibility to meningitis is affected.

As with the pGWAS, to improve our discovery power and mitigate false positives from cohort-specific batch effects, we performed similar associations in the other cohorts. The results for susceptibility to meningitis (MeninGene, Danish meningitis, UK biobank ICD-10 code for meningitis) found that position 74601544 on chromosome 15 (rs116264669) was associated with the minor allele increasing susceptibility in all three studies ($p = 4.4 \times 10^{-8}$; MAF = 3%) (Supplementary Figs. 10 and 11). This intronic SNP is located in *CCDC33*, a gene that has no prior

---

**Table 4 Human SNP heritability ($h^2_{SNP}$) of meningitis in the MeninGene cohort**

| Phenotype | Method | Heritability | Error | *p*-value |
|---|---|---|---|---|
| Susceptibility | GCTA | 0.25 | 0.05 | $2.4 \times 10^{-6}$ |
| (pneumococcal) | LDAK | 0.29 | 0.07 | $3.9 \times 10^{-6}$ |
| Severity (any | GCTA | 0.29 | 0.11 | $2.8 \times 10^{-5}$ |
| species) | LDAK | 0.49 | 0.14 | $1.4 \times 10^{-4}$ |

Heritability for susceptibility and severity of meningitis in Dutch adults. Heritabilities are shown on the liability scale (adjusted for population prevalence and case ascertainment ratio). We used two methods for each phenotype, GCTA and LDAK. The latter corrects for linkage disequilibrium when estimating the kinship between genotypes. All results showed significant ($p < 0.05$) evidence for a heritability above zero

---

**Table 5 Signals of human association in the MeninGene cohort**

| Phenotype Annotation | Position (SNP) | Marker | | Effect allele | MAF | OR | *p*-value | |
|---|---|---|---|---|---|---|---|---|
| Susceptibility (pneumococcal meningitis only) | chr6:117624549 | (rs210967) | G | 0.46 | 0.77 | $8.8 \times 10^{-7}$ | | *ROS1* intronic |
| | chr18:48403560 | (rs2850542) | T | 0.43 | 0.65 | $7.6 \times 10^{-8}$ | | *ME2* promoter (2 kb upstream of TSS) |
| | chr22:47506160 | (rs13057743) | G | 0.33 | 0.74 | $5.5 \times 10^{-7}$ | | *TBC1D22A* intronic |
| Severity (any species) | chr1:64680775 | (rs12081070) | A | 0.43 | 1.62 | $2.0 \times 10^{-8}$ | | *UBE2U* (fifth intron)/*ROR1* |
| | chr4:182823804 | (rs2309554) | A | 0.33 | 1.58 | $4.1 \times 10^{-7}$ | | *AC108142.1* intron (rs72739603) |
| A | chr9:37382231 | | | 0.07 | 2.36 | | $6.7 \times 10^{-7}$ | *ZCCHC7/GRHPR* |

We report the lead SNP at each putatively associated locus with MAF > 5% and $p < 1 \times 10^{-6}$, and nearby annotated genes. $p < 5 \times 10^{-8}$ is the genome-wide significance threshold—only rs12081070 exceeds this. The suggestive signal in all meningitis cases at rs3870369 was also present when restricted to pneumococcal cases, albeit with a lower $p$ value of $3.9 \times 10^{-7}$
MAF minor allele frequency, OR odds ratio, SNP single-nucleotide polymorphism, TSS transcription start site

association with infectious disease. *CCDC33* is expressed predominantly in the testes as well as the brain[47], although there is no evidence of an eQTL involving this variant or SNPs in linkage disequilibrium (LD) with it. The disease-associated variant is located in a genomic region that interacts with the immunoglobulin superfamily containing leucine-rich repeat 2 gene *ISLR2* on chromatin conformation capture analysis in macrophages (Supplementary Fig. 12A) and CD8 T cells (Supplementary Fig. 12B); moreover, a variant in complete LD (rs80140040) with the meningitis susceptibility SNP shows evidence of an eQTL with *ISLR2* in a number of tissues, including brain ($p = 0.02$). *ISLR2* shows the highest expression in the brain (neural tissues, Supplementary Fig. 12C) and plays a role in the development of the nervous system[54] but is poorly characterized in humans.

**Interactions between host and pathogen genomes**. It is possible that different host genotypes have varying susceptibility to different lineages or strains carrying certain alleles. This method of host–pathogen analysis has previously been applied to coding changes and host genotypes for human immunodeficiency virus (HIV)[55] and hepatitis C virus[56], but owing to their short genomes, these pathogens have many fewer variable sites than the pneumococcal population. To test for interactions, we used the 460 samples where we had collected both human genotype and pneumococcal genome sequence (Fig. 1). In the first instance, to retain hypothesis-free approach of GWAS, we performed a host–pathogen interaction analysis between every pair of common bacterial variants and genotyped host variants. While we were able to perform the $2 \times 10^{10}$ associations required, no pairs of loci surpassed the large multiple testing burden required by this analysis. A power calculation showed that we would have 80% power for finding an effect with MAF of 25% and OR of 4 (Supplementary Fig. 13) in the absence of population structure, which would only include strong and prevalent interaction effects.

This power analysis highlights the difficulty of reaching significance for this large number of tests with a relatively small number of samples. We then applied two further approaches in an attempt to find candidate interactions, with the caveat that due to the small sample number any results would require further validation to be conclusive. First, we considered regions with strong prior evidence for being involved in host–pathogen interaction. *S. pneumoniae* has many virulence factors, some of which are known to interact with specific human proteins[11]. We were interested in the interactions where the pneumococcal protein contains sequence variation, not totally confounded with genetic background. These regions have a higher power to be detected in an association analysis, and the higher rate of variation is potentially a sign of diversifying selection from interactions with the human immune system. We tested for an association between host genotype and the allele of three antigens selected for their variability and immunogenicity: PspC (CbpA), PspA, and ZmpA. For all of the antigen alleles with enough observations (Supplementary Table 7), we performed an association test against all imputed human variants as above, using a more accurate imputation of the *CFH* region due to its potential relevance in these interactions.

None of the bacterial antigen alleles were significantly correlated with variants in their human interacting-protein counterparts at a genome-wide significant or suggestive level ($p < 10^{-5}$). However, there were two associations of a *pspC* allele reaching genome-wide significance elsewhere in the genome. Supplementary Fig. 14 shows a LocusZoom plot of each of these associations. The first is between *pspC*−8 and position 148788006

on chromosome 6 (MAF = 0.08; OR = 9.20; $p = 4.1 \times 10^{-9}$). This is in *SASH1*, which has previously been found to have decreased expression during meningococcal meningitis (BioProject PRJNA106047). The second is between *pspC*−9 and position 98891272 on chromosome 13 (MAF = 0.16; OR = 6.30; $p = 3.6 \times 10^{-8}$) in *FARP1*, a gene not previously associated with infectious disease but expressed in the brain. We could find no published evidence of chromatin conformation capture interaction or eQTL effects with either of these human variants.

We then attempted to reduce the multiple testing burden by reducing the dimension of the pathogen genotype, which takes advantage of the extensive genome-wide correlation between variants. We used a genetic definition of pneumococcal lineages and tested whether pathogen genotypes, so defined, were associated with host genotype. We ran associations with lineages with at least 10% of samples in the subphenotype (Supplementary Table 8). The only result reaching genome-wide significance was an association between cluster eight (serotypes 9N/15B/19A, which have no overall association with invasive disease over carriage) and variants on chromosome 10 (Supplementary Fig. 15). The lead variant (rs10870273) is at position 134046136 on chromosome 10 (MAF = 0.27; OR = 4.28; $p = 1.2 \times 10^{-8}$) located in an intron of *STK32C*, a serine/threonine kinase highly expressed in the brain, which has also been shown to affect viral replication[57,58]. The high effect sizes estimated for the interactions are consistent with our power calculation (Supplementary Fig. 13).

## Discussion

Research on the role of pneumococcal variation in invasive potential in large epidemiological studies has mostly been focused on serotype variation. The lack of cohorts with whole genomes and invasive phenotypes has not permitted determination of other virulence factors in human disease—it is only with large cohorts of whole-genome sequences that contributions to pneumococcal phenotypes can be systematically attributed to serotype or other genetic variation. With our large collection of genomes, we were able to determine that the bacterial genome is crucial in determining invasive potential, and while serotype is likely to be an important factor, it alone was unable to account for the entire genetic contribution we estimated (maximum 77% of heritability explained). We went on to perform a combined analysis using 5892 pneumococcal genomes from two independent cohorts to find specific variation associated with invasive disease. We found genes independent of genetic background and serotype to be associated with invasive disease. This demonstrated a possible role for the virulence genes *pspC* (*cbpA*), *dacB*, and *psrP* in human disease and an association with the loss of the immunogenic protein ZmpD, as well as other proteins not known to be immunogenic. As it is impractical to cover all serotypes with the vaccine, conserved protein targets are needed to improve coverage and prevent serotype replacement[3]. The genes discovered here have potential as candidates that would block invasive disease in a serotype-independent manner, though further validation of their role in invasive disease would first be required.

Host genetics explained 29% ± 7% of variation in susceptibility to meningitis. As blood-borne pathogen invasion is assumed to be the route of infection in meningitis, we pooled samples from meningitis and bacteremia cases to undertake a better-powered hGWAS of IPD. We found a possible association at *CCDC33*. This gene does not have a known function that is related to immunity but functional genomic data suggest a possible mechanism for the susceptibility-associated variant through interacting at a distance with and modulating expression of the brain-expressed leucine-rich repeat and immunoglobulin (LIG)

family protein gene *ISLR2*. We found no evidence for pathogen genetics affecting the severity of disease, whereas human genetics explained 49% ± 9% of this variation. In our Dutch cohort, variation near *UBE2U* and *ROR1* was significantly associated with severity.

The MeninGene cohort also allowed a joint analysis of bacterial and human sequencing data. The high dimension of interaction data necessitates more samples than we were able to collect to find interaction effects of modest effect sizes, but through biologically guided dimension reduction we were able to show possible evidence for enrichment of certain pathogen genotypes in certain host genotypes. Owing to the approach of testing only specific candidates to reduce multiple testing burden, these results should be seen as speculative and as possible targets of future validation attempts.

Systematic surveys of genomic variation affecting infectious disease susceptibility and progression are a complementary approach for studying pathogenesis. The complexity and cost of an appropriate laboratory-based model negates the possibility of studying the effect of all host and pathogen genes on virulence using traditional wet-laboratory approaches. By combining large sequenced cohorts with clinical metadata, we were able to test for an effect of all observed sequence variants in both the host and pathogen, in the natural host. While finding associated human variants proved relatively challenging, as has been the case with other infectious diseases, our pGWASs found serotype-independent genes potentially involved in invasiveness. The inherent advantages of GWASs make joint analysis of human and bacterial genetics in other studies of infectious disease a useful approach, especially if both genomes can be collected concurrently.

## Methods

**Ethics statement.** For the MeninGene study, written informed consent was obtained from all patients or their legally authorized representatives. The study was approved by the Medical Ethics Committee of the Academic Medical Center, Amsterdam, The Netherlands (approval number: NL43784.018.13). For bacterial carriage samples from The Netherlands, written informed consent was obtained from both parents of each child participant and from all adult participants. Approvals for the 2009 and 2012/2013 studies in children and their parents (NL24116 and NL40288/NTR3614) and for the study in elderly adults (NTR3386) were received from a National Ethics Committee in The Netherlands (CCMO and METC Noord-Holland). For the 2010/2011 study, a National Ethics Committee in The Netherlands (the STEG-METC, Almere) decided that approval was not necessary. The studies were conducted in accordance with the European Statements for Good Clinical Practice and the Declaration of Helsinki of the World Medical Association. The Danish Invasive Pneumococcal Disease Cohort was approved by Danish Data Protection Agency (record no. 2007–41–0229 and 01864 HVH-2012–046). Ethical permission was obtained from The Ethical Committee of The Capital Region of Denmark (H-B-2007–085 and H-1–2012–063). According to Danish Legislation, the Research Ethics Committee can grant an exemption from obtaining informed consent for research projects based on biological material under certain circumstances, and for this study such an exemption was granted.

**Cohorts collected.** For MeninGene, we prospectively included patients of ≥16 years with CSF culture-proven bacterial meningitis between March 2006 and July 2015. Patients were identified from the database of The Netherlands Reference Laboratory for Bacterial Meningitis (NRLBM), which receives bacterial strains cultured from CSF and blood of 85% of community-acquired bacterial meningitis patients in The Netherlands. The NRLBM provided daily updates of hospitals in which patients with bacterial meningitis had been admitted in the preceding 2–6 days and names of treating physicians. Physicians were contacted by telephone by the investigators, informed about the study and asked to include the patient in the study. Physicians could also contact the investigators to include a patient without a report of the reference laboratory. Patients or their legal representatives were provided written information on the study and asked for written informed consent. Baseline, admission, treatment and outcome data was collected by the treating physician using an online case record form. Outcome was scored using the Glasgow Outcome Scale score, a five-point scale ranging from 1 (death) to 5 (mild or no disability).

Patients with hospital-acquired bacterial meningitis, defined as occurring during or within 1 week of hospital admission, were excluded as were those with a neurosurgical intervention or severe neurotrauma within 1 month before presentation and those with neurosurgical devices. All Dutch neurologists received information about MeninGene before and during the study.

Patient blood was collected in sodium/EDTA tubes and sent to the Academic Medical Centre, Amsterdam for DNA extraction. DNA was isolated with the Gentra Puregene Isolation Kit (Qiagen), and quality control procedures were performed to determine the yield and purity.

Pathogens were stored at −80 °C in the NRLBM upon receipt. Isolates were recultured from frozen stock on blood agar plates, from which DNA was extracted directly. Sequencing was performed using multiplexed libraries on the Illumina HiSeq platform to produce paired end reads of 100 nucleotides in length (Illumina, San Diego, CA, USA).

Controls for the Dutch meningitis susceptibility GWAS and heritability analysis have been composed of the healthy adult controls from the amyotrophic lateral sclerosis (ALS) and B-Vitamins for the PRevention Of Osteoporotic Fractures (B-PROOF) cohorts[59,60].

For the Danish Childhood Pneumococcal Disease Cohort, cases of IPD in children aged <5 years were identified through the National Neisseria and Streptococcus Reference Laboratory, Statens Serum Institut (SSI), Copenhagen, Denmark[61]. DNA was obtained from the Danish Neonatal Screening Biobank (DNSB), SSI. Cases with a prior hospitalization for any cause were excluded in order to exclude children with severe comorbidity. Information on hospitalization was obtained from the Danish National Patient Register. Genomic DNA was extracted from dried blood spots using the Extract-N-Amp Kit (Sigma-Aldrich) and the REPLI-g Mini Kit (Qiagen). Cases were genotyped on the Illumina HiScan platform with the Human Omni1-Quad beadchip. As population-matched controls for the Danish collection, we used the genotypes of 2805 randomly sampled healthy Danish young adults from the GOYA study (Genomics of extremely Overweight Young Adults)[62].

For Genetics of Sepsis and Septic Shock in Europe (GenOSept), cases were defined as patients with confirmed septic shock managed in an intensive care environment recruited as part of the GenOSept consortium with a diagnosis of pneumococcal infection confirmed using either blood cultures positive for *S. pneumoniae* or positive pneumococcal urinary antigen. Population controls were derived from the Wellcome Trust Case Control Consortium 2 1958 Birth Cohort (dbGaP accession phs900028.v1.p1). The GenOSept cases were genotyped on the Affymetrix 5.0 and the WTCCC2 controls on the Affymetrix 6.0 arrays and were quality checked in parallel using the following steps. After mapping SNPs to build 37 coordinates samples with call rates either <98% or those with mismatching reported and genetically defined sex or with relatedness IBD score >0.2 were removed. SNPs with call rates <98% or Hardy–Weinberg equilibrium (HWE) $p < 1 \times 10^{-10}$ or MAF < 1% were also removed. Finally, multidimensional scaling (MDS) was used to define sample ancestries and any outliers based on either MDS or heterozygosity (>3 standard deviations around the mean) were removed. Intersecting SNPs between the two arrays were then used as a scaffold for variant imputation that was performed simultaneously using SHAPEIT and IMPUTE2.2 using the 1000 Genomes phase 1 variant set release. Default settings were used throughout phasing and imputation steps. Association testing of imputed probabilities was performed using SNPTEST version 2.4.0 using additive frequentist methods tests using the first 4 MDS components as covariates. Summary statistics were obtained from this susceptibility GWAS of severe pneumococcal infection[63].

From UK Biobank, we extracted case samples with self-reported meningitis or sepsis/septicaemia (data-field 20002), with a diagnosis of meningitis (data-field 41202 having a value of G01, G001, G002, G003, G008, A170, A390 or A321 at least once) and with a diagnosis of sepsis (A403, A409, A408 or A40 at least once). We randomly selected 3000 control samples from the remaining samples that had passed the UK Biobank's genetic quality control step, which allowed for quicker analysis with little impact on effective sample size.

Summary statistics were obtained from a GWAS of bacterial meningitis performed by 23andme, with cases defined by those who responded yes to the question 'Have you ever had bacterial meningitis?'[8].

Carriage of *S. pneumoniae* in Dutch adults and children was determined by conventional culture in vaccinated children (11 and 24 months of age) and their parents/adults in 2009[64], 2010/2011[64] and 2012/2013[65] and in community-dwelling elderly adults in 2011–2013[66]. All children were vaccinated with PCV-7 or PHiD-CV10 according to the Dutch national immunization program at 2, 3, 4 and 11 months of age. Nasopharyngeal swabs were collected from all individuals and oropharyngeal swabs were collected from all adult subjects by trained study personnel using flexible, sterile swabs according to the standard procedures described by the World Health Organization[67]. After sampling, swabs were immediately placed in liquid Amies transport medium, transported to the microbiology laboratory at room temperature and cultured within 12 h. From elderly participants, saliva was also collected using Oracol Saliva Collection System (Malvern Medical Developments Ltd, Worcester, UK), immediately transferred to tubes pre-filled with glycerol and transported to the diagnostic laboratory on dry ice[66]. After being cultured for *S. pneumoniae*, samples from adults were tested for pneumococci with molecular diagnostic methods[66,68]. Samples identified as positive with molecular methods, yet negative when cultured at primary diagnostic step were revisited with culture in the second attempt to isolate live pneumococci[69]. Pneumococcal isolates were identified using conventional methods; one pneumococcal colony per plate was subcultured (more if distinct morphotypes

were observed) and serotyped by capsular swelling (Quellung). DNA extraction and sequencing was performed as for the MeninGene study.

In the South African cohort, pneumococcal isolates from disease were identified in a national, active, laboratory-based surveillance for IPD—defined as isolation from a normally sterile site—by the National Institute for Communicable Disease, Johannesburg between 2005 and 2014[70]. Pneumococcal isolates from asymptomatic carriage were collected in two cross-sectional colonization surveys in Agincourt and Soweto between 2009 and 2013[71]. Isolates from these collections were randomly subsampled for sequencing on Illumina HiSeq (as for the MeninGene study) within the age groups ≤2 (50%), >2–≤5 (25%) and >5 (25%). Serotyping was performed using Quellung and DNA was extracted from single colony subcultures from stocks stored at −80 °C.

**Cataloguing bacterial variation.** From the whole-genome sequence data of bacteria in the cohort, we called SNPs and short INDELs with respect to the ATCC 700669 reference[72]. We mapped reads with bwa mem[73] and marked optical duplicates with Picard. Using this mapping, we used cn.mops[74] to call copy number variations (CNVs). We used windows of 1 kb and used windows with support for a CNV in at least two samples. The number of reads mapping to each *ivr* allele in each sample was determined by counting correctly oriented mapped read pairs spanning the repeats in the locus[16,75].

SNPs and INDELs were then called with GATK HaplotypeCaller[76]. For INDELs, we used the recommended hard filters. For SNPs, we used the recommended hard filters to create an initial call set. We then applied GATK VariantRecalibrator using the following call sites as true positive priors: the intersection of SNPs called by both GATK and bcftools (Q10; 90% confidence); filtered SNPs from a carriage cohort of Karen infants[77] (Q5; 68% confidence); filtered SNPs from a carriage cohort of children in Massachusetts[36] (Q5; 68% confidence). After quality score recalibration, we used 99.9% recall as a cutoff for SNPs to maximize sensitivity and annotated the predicted effect of all coding variation using the variant effect predictor[78]. We defined LoF variants as either stop gained or frameshift mutations. We used PROVEAN with a score cutoff of < −2.5 to predict whether non-synonymous SNPs affect protein function[79].

We counted variable length k-mers with a minor allele count of at least ten using fsm-lite[18]. In the Dutch data, there were 11.7M informative k-mers, with 2.6M unique patterns. While the k-mer approach should directly assay or tag most variation at the population level, the allelic variation of the pneumococcal antigens may not be well captured. For example, *pspC* can be difficult to assemble due to repeats and CNV[80], and k-mers from *pspA* and *zmpA* may not map to each allele specifically due to orthologous and paralogous genes[35,81]. We developed a more accurate way to classify the allele present in each isolate combining assembly and mapping statistics. For each antigen in each sample, we mapped reads to a reference panel using srst2[82] and aligned annotated genes from the assemblies using blastp. We then built a classifier using the summary statistics reported by these programs. For *pspC*, we used an existing classification scheme of 11 alleles from 48 sequences as training data (Supplementary Fig. 16)[80]. For *zmpA* and *pspA*, we built trees from previously characterized alleles[27] (Supplementary Figs. 17 and 18), which allowed us to define four allele groups for *pspA* and three for *zmpA*. We ensured that the unlabelled training data were separable into these groups using principal component analysis (PCA; Supplementary Fig. 19). We tested the performance of four out-of-the-box classifiers on 20 *pspC* alleles spread across the population that we manually typed from the assemblies (Supplementary Table 9). Finding that a support vector machine with a linear kernel worked best, we used this to classify the allele of all antigens in all isolates using the summary statistics described above (Supplementary Fig. 20).

Using the South African samples, we counted k-mers, SNPs and INDELs and COGs in the same way and annotated LoF function variants. We found 52,215 SNPs and INDELs, 6.3M informative k-mers, with 1.5M unique patterns.

**Association of bacterial variation.** Using the SNP and INDEL alignment, we built a phylogenetic tree from this alignment using fasttree[83] and calculated the kinship (covariance matrix) between each pair of strains as the distance between their MRCA and the midpoint root[84]. All heritabilities and variance components were calculated with the limix package (v2.0.0)[85]. We calculated the heritability of each phenotype using this kinship matrix and Bernoulli errors. To estimate the heritability due to serotype alone, we performed the same calculation but with the kinship matrix calculated from a genotype matrix that treats each possible serotype as a genetic variant. Owing to the relative sparsity of serotype observations, it was not possible to do this analysis for individual phenotypes. We repeated both of these analyses while using the top ten principal components of a PCA of the genotype matrix (67% variance explained) as covariates. Finally, we performed variance decomposition using these two kinship matrices to split the detected heritability into components due to serotype, other genetics and residuals[86]. This required treating invasiveness as a continuous phenotype. We also tested whether the effect of serotype on invasiveness could be explained by capsule charge on phenotype, by using previously measured zeta potentials in place of serotype[87], using the serogroup average when a serotype-specific charge was not known. Invasiveness was not well predicted from capsule charge alone ($R^2 = 0.08$)[88], partly because of the unknown capsule charge for many of the serotypes observed.

For association of common variation (MAF > 1%), we compared SEER[18], using the first ten MDS components as fixed effects to control for population structure, with FaST-LMM[85], which uses eigenvectors from the kinship matrix calculated from the SNP and INDEL alignment as random effects to control for population structure. The Q-Q plots using fixed effects were highly inflated (Supplementary Fig. 21), so we used the linear mixed model throughout. To correct for multiple testing, we used the number of unique patterns as the number of tests in a Bonferroni correction, giving $p < 8.2 \times 10^{-7}$ for SNPs and $p < 1.9 \times 10^{-8}$ for k-mers. Inspection of the Q-Q plots showed inflation of the test statistic for k-mers, so we used a higher threshold of $p < 10^{-16}$ instead. The same association methods was used with the antigen alleles and CNVs.

We considered whether the *ivr* locus, a phase variable inverting type I R-M system with six possible alleles[75], is associated with meningitis, as has been previously shown using mouse models of disease[29,88]. The rapid variation of this locus allows simple associations independent of genetic background. To test for association of the *ivr* locus alleles with susceptibility and severity, we used a Bayesian hierarchical model to find differences in the proportion of alleles present in tissue types while accounting for heterogeneity within single colony picks[16]. We found no evidence that either allele frequencies or overall diversity had any association with invasive disease or carriage in clinical cases of meningitis (Supplementary Figs. 22 and 23).

We calculated Tajima's *D* for all coding sequences annotated in the ATCC 700669 reference, ignoring gaps or unknown sites and calculated *p* values for difference between niche using null permutations of phenotype labels[89]. We applied a Bonferroni correction using the number of coding sequences tested multiplied by the number of association techniques (three), with the number of null permutations (44,000) chosen allowing for detection of significance at this adjusted threshold.

Rare variants (MAF < 1%) were associated by grouping variants by coding sequence in a burden test, in which we treated samples with a mutation anywhere in the gene as the alternate allele and unmutated genes as the reference allele[90]. As these burden tests lose power when variants have different directions of effect on the phenotype, we used only those variants predicted to cause a LoF in one test, and those causing either LoF (6825 variants) or predicted change in protein function in another (additional 26,206 variants). We used plink/seq to perform this association for each phenotype, applying a Bonferroni correction using the number of genes as the number of multiple tests ('burden of rare LoF/missense' in Table 1). We then repeated this using the LMM burden testing mode of pyseer[91], which allowed us to correct for population structure with the same model as above. Reasoning that power to detect individual variants may also be hampered by population structure as well as allele frequency, we also searched for a burden of missense variants of any frequency by gene ('missense burden' in Table 1), though this test would be insensitive to genes where missense variants have different directions of effect. For this test, we classified samples containing a missense variant anywhere in the gene as the alternative allele. We excluded sequences which appeared to be pseudogenes in the population, as the variant annotation is no longer likely to be meaningful. To attempt to calculate heritability due to rare variation, we used these definitions of burden to form a genotype matrix. We were unable to find significant evidence for $h^2_{burden} > 0$ for any individual gene or over all genes. We were also unable to separate out this variance component from the overall estimated heritability from core SNPs $h^2_{SNP}$. Rather than ruling out a role for rare variation in pathogenesis, this is instead likely due to a combination of methodological factors: a small genotype matrix limited by the number of genes, rarity of mutations even when aggregated, and potentially grouping variants with different directions of effect on invasiveness.

When performing meta-analysis with both the Dutch and South African samples, we pooled the genetic data and performed the same association analysis as for the Dutch data alone. Where possible, we included country as a covariate. The South African cohort also included host gender, age, collection year and HIV status and PCV use at the time of sampling. We included these as additional covariates for these samples. We performed a burden test using only damaging LoF variants. We used a significance threshold of $p < 0.05$ in the pooled analysis, after applying a Bonferroni correction for multiple testing based on the number of unique patterns as above. For all tests, we ensured that the Q-Q plots of the resulting *p* values were not inflated (Supplementary Fig. 24).

**Human genotyping and quality control.** We performed genotyping using the Illumina Omni array and called genotypes from normalized intensity data using optiCall[92]. For data taken from other platforms, we merged cases and controls only at sites in the intersection of the genotyping arrays used. We then performed basic quality control steps to first remove low quality samples, then low quality markers[93]. Samples with a heterozygosity rate three standard deviations away from the mean or >3% missing genotypes were removed. Markers with >5% missing genotypes, significantly different ($p < 10^{-5}$) call rate between cases and controls, MAF < 1% or out of HWE ($p < 10^{-5}$) were removed. Using an LD-pruned set of markers, we estimated sample relatedness with KING[94] and removed any duplicate samples. Using the same set of markers, we used eigenstrat to perform a PCA to check sample ancestry (Supplementary Fig. 25)[95]. Samples closer than third-degree relation and samples of non-European ancestry (which we defined as PC1 < 0.07) were removed for heritability analysis. We manually inspected intensity plots for

any associated markers using Evoker[96] and removed any miscalled sites. Finally, we removed markers significantly associated with control batch ($p < 5 \times 10^{-8}$).

All markers were reported with respect to the reference allele and coordinates of GRCh37. We imputed markers using the HRC as a reference panel with the Sanger Imputation Server[97–99]. For the Danish samples, we instead used the Michigan imputation server due to the decreased number of markers available from merging two different genotyping arrays[100]. For greater accuracy, the *CFH* region was imputed using impute2 with 1000 Genomes and GoNL as reference panels[101–103]. We removed resulting markers with MAF < 1%, HWE $p < 10^{-5}$ or INFO scores <0.7 leaving 6.8M markers for association testing and heritability estimation.

**Association of human variation**. Throughout we used a Glasgow Outcome Scale score[104] of <5 to define unfavourable outcome, as described previously[4]. We performed the association study using bolt-lmm[44,105], using the LD-pruned set of genotyped markers to estimate the kinship matrix, and then calculating association statistics for all genotyped and imputed sites passing the above quality control thresholds. For the Dutch samples, we included whether the patient was immunocompromised as a fixed effect (10% of cases), assuming that no control samples were immunocompromised (1% population prevalence[106,107]). To estimate heritability, we used two methods: GCTA-GREML[108] (as implemented in bolt-lmm) and LDAK v5[109]. For both, we only used samples passing the stricter thresholds for ancestry and relatedness. Estimates of heritability were transformed from the observed scale to the liability scale using a population prevalence of meningitis of $1 \times 10^{-3}$.

To perform associations using the UK Biobank, we used bolt-lmm, following the recommended protocol for analysing the available genetic data[110]. Using the sample described above, we removed genotyped markers with MAF < 0.001 or a missing rate >0.1 and used this to estimate kinships in bolt-lmm. We used bolt-lmm to perform association analysis of every imputed SNP site, including participant age as a fixed effect.

We used METAL to perform meta-analysis between different sets of studies[111]. We used the effective sample size to weight the beta and SE from each set of summary statistics, also adjusting the beta values and standard errors produced by bolt-lmm (Supplementary Table 10). We only retained markers that had been successfully imputed in all studies, to avoid effects of varying sample size at each locus.

Chromatin conformation capture data were tested and presented using the Capture HiC Plotter[112] and eQTL data from the GTEx Consortium[47].

**Interaction effects**. We took 460 pneumococcal meningitis samples with matched pathogen and human sequence data that passed quality control thresholds for both data types. To test all variants in a pairwise manner, we treated human genotypes as additive and stored site and sample data separately for more efficient access by chunk[113]. The number of pairwise tests between all common variants was prohibitively large ($10^{12}$), so we only tested genotyped markers: $1.8 \times 10^{10}$ pairs of variants passed filters of MAF > 5% and missing rate <5% in both the human and pathogen data. We modified the association code of SEER[18] to extend the $\chi^2$ test to a $3 \times 2$ table and to perform a $3 \times 2$ Fisher's exact test when assumptions of the $\chi^2$ test were violated. Those sites with $p < 5 \times 10^{-11}$ (a Bonferroni correction with $\alpha = 1$, as an initial filter) were then tested using a logistic regression of the human SNP against the pathogen variant, with the first three components from MDS of the pathogen kinship matrix included as covariates to adjust for pathogen population structure.

To test for an association between invading lineage and human genotype, we ran hierBAPS on these 460 samples[114], which generated ten top-level clusters, seven of which were large enough to test (Supplementary Table 8). To perform the association, we used bolt-lmm, as above. We tested association of antigen alleles with frequencies >10% in the sampled population in the same way (Supplementary Table 7).

## Data availability

Bacterial metadata, including ENA accession numbers for sequencing data, can be accessed on figshare 5915314 [https://doi.org/10.6084/m9.figshare.5915314]. Summary statistics from the pGWAS can be found on figshare 7728620 [https://doi.org/10.6084/m9.figshare.7728620]. Summary statistics from the hGWAS can be found on Zenodo 2572916 [https://doi.org/10.5281/zenodo.2572916].

## Code availability

The code used for the analysis, along with phylogenies and predicted antigen alleles, can be found at https://github.com/johnlees/meningene (BSD-3 License). In addition, the following software packages were used: bwa mem (v0.7.10); Picard (v1.124); cn.mops (v1.28.0); GATK (v3.5.0); Provean (v1.1.5); srst2 (v0.2.0); fasttree (v2.1.8); limix (v2.0.0); SEER (v1.4.0); FaST-LMM (python) (v0.2.32); plink/seq (v0.09); pyseer (v1.0.2); opticall (v0.7.0); KING (v1.4); eigenstrat (v10210); Evoker (v2.2); bolt-lmm (v2.3); LDAK (v5.0); METAL (v2011–03–25); hierBAPS (v6.0).

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

## Acknowledgements

Win Kit Man isolated genomic DNA from bacteria in the Dutch cohort. We would like to thank Chao Tan and David Hinds from 23andme for sharing summary statistics from their association with self-reported bacterial meningitis. We also thank all members of the GenOSept consortium for making their data available and Anna Rautanen in particular for her preparation of the genetic data. This research has been conducted using the UK Biobank Resource under Application Number 25747. This study also makes use of data generated by the Genome of The Netherlands Project (GoNL). Funding for the GoNL project was provided by The Netherlands Organization for Scientific Research under award number 184021007, dated July 9, 2009 and made available as a Rainbow Project of the Biobanking and Biomolecular Research Infrastructure Netherlands (BBMRI-NL). Work at the Wellcome Trust Sanger Institute was supported by Wellcome (098051). This work was also supported by grants from the European Research Council (ERC Starting Grant, proposal/contract 281156; https://erc.europa.eu/) to D.v.d.B. and the Netherlands Organization for Health Research and Development (ZonMw; NWO-Vidi grant 2010, proposal/contract 016.116.358 to D.v.d.B.; NWO-Veni Grant NWO-Veni grant 2012 proposal/contract 916.13.078 to M.B.; www.zonmw.nl/). The Netherlands Reference Laboratory for Bacterial Meningitis was supported by the National Institute for Health and Environmental Protection, Bilthoven (www.rivm.nl/). J.A.L. was supported by a Medical Research Council studentship grant (1365620). N.J.C. was supported by a Sir Henry Dale Fellowship, jointly funded by the Wellcome Trust and the Royal Society (grant number 104169/Z/14/Z). J.N.W. is funded by grants from the United States Public Health Service (AI038446 and AI105168). The B-PROOF cohort was funded by The Netherlands Organization for Health Research and Development (ZonMw, Grant 6130.0031), the Hague; unrestricted grant from NZO (Dutch Dairy Association), Zoetermeer; Orthica, Almere; NCHA (Netherlands Consortium Healthy Ageing) Leiden/Rotterdam; Ministry of Economic Affairs, Agriculture and Innovation (project KB-15–004–003), the Hague; Wageningen University, Wageningen; VUmc, Amsterdam; Erasmus Medical Center, Rotterdam; Unilever, Colworth, UK. Dutch Carriage studies at RIVM were funded by the Dutch Ministry of Health, Welfare and Sport. Genotyping for GOYA was funded by the Wellcome Trust (WT 084762MA). GOYA is a nested study within The Danish National Birth Cohort, which was established with major funding from the Danish National Research Foundation. Additional support for this cohort has been obtained from the Pharmacy Foundation, the Egmont Foundation, The March of Dimes Birth Defects Foundation, the Augustinus Foundation, and the Health Foundation. Funding for the Danish Invasive Pneumococcal Disease Cohort was kindly provided by the Lundbeck Foundation, the Novo Nordisk Foundation, King Christian the 10th Foundation, Jacob Madsen's Foundation, TrygVesta, Ebba Celinder's Foundation, the Danish Medical Association Foundation, the Foundation for Advancement of Medical Science, the Augustinus Foundation, Peder Laurits Pedersen's Foundation, A.P. Møller Foundation for the Advancement of Medical Science, the Danish Medical Research Council, Preben and Anna Simonsen's Foundation, Ferdinand and Ellen Hindsgaul's Foundation, the Hartmann's Foundation and Dagmar Marshalls Fond. The GenOSept study was supported by the European Union (6th framework programme of RTD funding) and the National Institute for Health Research, through the Comprehensive Clinical Research Network. J.C.K. was supported by NIHR Oxford Biomedical Research Centre and a Wellcome Trust Investigator Award (204969/Z/16/Z) and A.J.W.-M. was supported by a Wellcome Trust Fellowship with reference 106289/Z/14/Z. The funders had no role in study design, data collection and analysis, decision to publish, or preparation of the manuscript.

## Author contributions

Conceptualization: J.A.L., B.F., P.H.C.K., J.P., A.v.G., A.v.d.E., M.C.B., J.C.B., S.D.B., D.v.d.B. Data curation: J.A.L., B.F., P.H.C.K., N.E.W., N.J.C., R.A.G., H.J.B., Z.B.H., L.H.Ä., A.J.W.-M., T.C.M. Formal analysis: J.A.L., B.F., P.H.C.K., N.E.W., A.H.Z., A.J.W.-M., J.C.K. Funding acquisition: J.P., S.M., T.B., A.v.G., A.v.d.E., M.C.B., J.C.B., S.D.B., D.v.d.B. Investigation: J.A.L., B.F., P.H.C.K., N.E.W., M.V.S., A.J.W.-M., T.C.M., J.C.K. Methodology: J.A.L., B.F., P.H.C.K., N.E.W. Project administration: A.v.d.E., M.C.B., J.C.B., S.D.B., D.v.d.B. Resources: M.V.S., H.J.B., N.Y.R., A.J.W.-M., E.A.M.S., K.T., A.L.W., L.H.v.d.B., W.v.R., J.H.V., Z.B.H., L.F.L., L.C.P.G.M.d.G., N.M.v.S., L.H.Ä., T.I.A.S., E.A.N., M.d.P. Software: J.A.L. Supervision: J.N.W., J.P., S.M., T.B., A.v.G., A.v.d.E., M.C.B., J.C.B., S.D.B., D.v.d.B. Writing—original draft: J.A.L., S.D.B., D.v.d.B. Writing—review and editing: all authors.

## Additional information

**Competing interests:** N.J.C. and S.D.B. were consultants for Antigen Discovery, Inc involved in the design of a proteome array for *S. pneumoniae*. E.A.M.S. reports grants from the pharmaceutical companies GlaxoSmithKline and Pfizer outside the submitted work. K.T. reports grants from Pfizer and consultancy fees from Pfizer paid to University

Medical Centre Utrecht, both received outside the submitted work. A.L.W. received consulting fees for participation in advisory boards for Pfizer, outside the submitted work. D.v.d.B. received departmental honoraria for serving on a scientific advisory board for GlaxoSmithKline and InflaRx paid to the Amsterdam UMC, outside the submitted work. All the other authors declare no competing interests.

John A. Lees[1,2], Bart Ferwerda[3], Philip H.C. Kremer[3], Nicole E. Wheeler[2,4], Mercedes Valls Serón[3], Nicholas J. Croucher[5], Rebecca A. Gladstone[2], Hester J. Bootsma[6], Nynke Y. Rots[6], Alienke J. Wijmega-Monsuur[6], Elisabeth A.M. Sanders[6,7], Krzysztof Trzciński[7], Anne L. Wyllie[7,8], Aeilko H. Zwinderman[9], Leonard H. van den Berg[10], Wouter van Rheenen[10], Jan H. Veldink[10], Zitta B. Harboe[11], Lene F. Lundbo[12], Lisette C.P.G.M. de Groot[13], Natasja M. van Schoor[14], Nathalie van der Velde[15,16], Lars H. Ängquist[17], Thorkild I.A. Sørensen[18,19], Ellen A. Nohr[20], Alexander J. Mentzer[21], Tara C. Mills[21], Julian C. Knight[21], Mignon du Plessis[22], Susan Nzenze[22], Jeffrey N. Weiser[1], Julian Parkhill[2], Shabir Madhi[23], Thomas Benfield[12], Anne von Gottberg[22,23], Arie van der Ende[24,25], Matthijs C. Brouwer[3], Jeffrey C. Barrett[2,26], Stephen D. Bentley[2,27] & Diederik van de Beek[3,27]

[1]Department of Microbiology, New York University School of Medicine, New York, NY 10016, USA. [2]Parasites and Microbes, Wellcome Sanger Institute, Cambridge CB10 1SA, UK. [3]Amsterdam UMC, University of Amsterdam, Department of Neurology, Amsterdam Neuroscience, Meibergdreef 9, Amsterdam 1105 AZ, The Netherlands. [4]The Centre for Genomic Pathogen Surveillance, Wellcome Sanger Institute, Cambridge CB10 1SA, UK. [5]MRC Centre for Global Infectious Disease Analysis, Department of Infectious Disease Epidemiology, Imperial College London, London W2 1PG, UK. [6]Centre for Infectious Disease Control, National Institute for Public Health and the Environment, Bilthoven 3721 MA, The Netherlands. [7]Department of Pediatric Immunology and Infectious Diseases, Wilhelmina Children's Hospital, University Medical Centre Utrecht, Utrecht 3508 AB, The Netherlands. [8]Epidemiology of Microbial Diseases, Yale School of Public Health, New Haven, CT 06520, USA. [9]Amsterdam UMC, University of Amsterdam, Department of Clinical Epidemiology, Biostatistics and Bioinformatics, Amsterdam Public Health, Meibergdreef 9, Amsterdam 1105 AZ, The Netherlands. [10]Department of Neurology, Brain Center Rudolf Magnus, University Medical Center Utrecht, Utrecht 3584 CG, The Netherlands. [11]Department of Microbiological Surveillance and Research, Statens Serum Institut, Copenhagen DK-2300, Denmark. [12]Department of Infectious Diseases, Hvidovre Hospital, University of Copenhagen, Hvidovre 2650, Denmark. [13]Department of Human Nutrition, Wageningen University, P.O. Box 176700 AA Wageningen, The Netherlands. [14]Amsterdam UMC, VU University, Department of Epidemiology and Biostatistics, Amsterdam Public Health, Van der Boechorststraat 7, Amsterdam 1007 MB, The Netherlands. [15]Amsterdam UMC, University of Amsterdam, Department of Internal Medicine, Geriatrics, Amsterdam Public Health, Meibergdreef 9, Amsterdam 1105 AZ, The Netherlands. [16]Department of Internal Medicine, Erasmus MC, University Medical Centre Rotterdam, P.O. Box 20403000 CA Rotterdam, The Netherlands. [17]Center for Clinical Research and Disease Prevention, Bispebjerg and Frederiksberg Hospitals, The Capital Region, Copenhagen DK-2000, Denmark. [18]The Novo Nordisk Foundation Center for Basic Metabolic Research, Section of Metabolic Genetics, Copenhagen DK-2200, Denmark. [19]The Department of Public Health, Section of Epidemiology, Faculty of Health and Medical Sciences, University of Copenhagen, Copenhagen DK-1014, Denmark. [20]Institute of Clinical Research, University of Southern Denmark, Odense DK-5000, Denmark. [21]Wellcome Centre for Human Genetics, Nuffield Department of Medicine, University of Oxford, Oxford OX3 7BN, UK. [22]School of Pathology, Faculty of Health Sciences, University of Witwatersrand, Johannesburg 2000, South Africa. [23]National Institute for Communicable Diseases, Johannesburg 2192, South Africa. [24]Amsterdam UMC, University of Amsterdam, Department of Medical Microbiology, Amsterdam Infection and Immunity, Meibergdreef 9, Amsterdam 1105 AZ, The Netherlands. [25]Netherlands Reference Laboratory for Bacterial Meningitis, Amsterdam UMC/RIVM, University of Amsterdam, Meibergdreef 9, Amsterdam 1105 AZ, The Netherlands. [26]Genomics Plc, East Road, Cambridge CB1 1BH, UK. [27]These authors contributed equally: Stephen D. Bentley, Diederik van de Beek.

