## [Peer Review File · Nature Communications]

Reviewer #1 (Remarks to the Author):

This is a very interesting manuscript presenting a thorough analysis of genetic variation of both host and pathogen affecting meningococcal disease, with an impressive array of statistical tests and that finds several interesting genes, especially on the bacterial side. It is also very commendable to try to look for interactions, although unsurprising that it is currently underpowered.

The manuscript is however somewhat weaker in biological interpretation. It is written in many ways like a standard human genetics GWAS manuscript but this is not quite appropriate because the audience is different and because there are several interpretational issues that are specific to bacterial GWAS especially because of the key issue of population structure.

(1) it is important to "see" the bacterial data. This can be done by trees which show the distribution of carriage and disease strains.

(2) the heritability of bacterial genotypes seems very high and this should be put in a bit more context. Is an additive model appropriate and interpretable? It is also worth spelling out what would a heritability of 1.0 mean. how much heritability could be explained by the top PCs (which might in part be due to differences between the populations that carriage and disease were sampled from)? The authors assume that serotypes are likely to be important but presumably their importance is highly confounded with population structure, like for most other genes. Is there any evidence for effect, individually or collectively of serotype when this is controlled for? As it is quoting the figure of 0.45 seems a bit misleading and not particularly relevant to overall interpretation.

(3) If the sampling has been done well, then the disease population should be thought of as a non-random subset of the carriage population (i.e. with specific genotypes amplified), rather than a separate population, since all disease isolates were carriage isolates before causing disease. Therefore difference in Tajima's D as well as the results on heritability should be discussed in this light. It seems from a brief reading that the controls are pretty good.

(4) It is notable that so many of the hits are for "missense burden", but this phrase is only ever used in the table, never in the methods and I would like to be more sure that I know exactly what was done here and also I would like this finding interpreted, both biologically and methodologically especially because of the very high p-values that have been obtained for rather obscure genes. One speculation is that because so much of the difference is confounded with phylogenetic (clonal) structure, the main GWAS signal that remains is rare mutations that happen repeatedly at the tips of the tree. Are the genes with the highest p-values particularly prone to missense variants?

(5) Relating to point 4, The results on different number of lof and damaging mutations between disease and carriage isolates is really intriguing and could hold a key to understanding disease

biology but this is never interpreted. It is even more so because some genes seem to go in the opposite direction (based on the missense table, if I am interpreting it correctly). It seems like there is a systematic difference in vigor between disease and carriage isolates, with disease isolates more vigorous. Presumably rare mutations are unlikely to be included in heritability, making the true genomic heritability higher than the estimate? What is the overall proportion of heritability that can be explained by aggregate LoF burden (with and without population structure control) and by individual genes. It would also be interesting to see the effect of LoF at different gene frequencies. Are very rare variants responsible for much of the the effect?

(6) The human genetics section is at present rather hard going for people who are not familiar with the nitty gritty of today's human GWAS. It is hard to get a sense of how believable exciting the findings are, especially in terms of how surprising/convincing the eQTL and chromatin conformation capture analysis results are in terms of enriching/validating the signal. This section could perhaps be rewritten with a slightly broader audience in mind, e.g. highlighting entry points to the literature for comparable scans.

Daniel Falush

Reviewer #2 (Remarks to the Author):

There is an opportunity to jointly assess the infecting agent and the host in GWAS analyses. This is the stated goal in this work where the authors perform genetic association studies for the pathogen, for host susceptibility, and a genome-to-genome analysis. The target pathogen is pneumococcus, and the main phenotypes are meningitis and invasive disease. By aggregating data, the authors reach some thousand isolate genomes, some thousand host genotypes, and a ~400 samples that are doubly typed. Unfortunately, there is limited attention to the immense heterogeneity that is brought to the analysis – including the fact that some of the cohorts are from different populations, several organisms, multiple possible pneumococcal populations, and an unknown number of confounders that are not controlled for.

There are a number of hesitations in methodology: some of the analyses combine various data sources without careful management of data structure (starting with host ancestry and pathogen clonal structure surveys). Some of the analyses would have been more appropriate as meta-analyses than as combined cohorts. In addition, lack of power pushes the work to slice and dice by nesting targeted, candidate analyses, within GWASs. This is generally not accepted by the genomic community except if there is strong downstream validation.

There is valuable data in this manuscript – but the current approach does not support the overall claims.

Specific comments:

1. Figure 2. I am surprised by the increase in most types of substitutions: including synonymous and intergenic. Could this be explained by more clonal nature of the invasive strains? It would be important to add a ratio analysis to the analysis of burden – where the LoF and damaging variants numbers are corrected by the synonymous (+intergenic) variation. From this analysis it is possible to surmise the excess burden. It would be useful to complement the Tajima's D test with a measure of selection such as dN/dS.

2. I am worried about unaccounted for confounding when combining different cohorts and different locations (eg. Dutch meningitis plus South Africa invasive pneumococci)- a less worrisome approach would be as metaanalysis. At least, the behavior per cohort should be presented for the genes in Table 1.

3. Table 1. The use of multiple methods can be problematic, because it increases multiple testing. It should be made clear how p values will be adjusted both for the hypothesis tested, the number of variants and the number of techniques employed.

4. Table 2, and elsewhere in the manuscript, there is analysis of "meningitis" – this is probably any type of meningitis, and not just pneumococcal. For the sake of consistency and clarity, the analysis of multi-cause meningitis should not be the object of analysis in this work. Severity should also be clarified – is it pneumococcal-proper, or all-cause? Such a heterogeneity could also result in spurious associations due to unaccounted for strata in the data.

5. There are several convenience analyses of other cohorts with various infectious disease causes and outcomes (Danish bacteremia, GenOSept and self-reported cases of sepsis in the UK biobank, 23andMe) that do not help with the message of the manuscript.

6. The lack of power moved the authors to concentrate genome-genome analysis to few selected genes and also to "to reduce the multiple testing burden by reducing the dimension of the pathogen genotype". In practice, these approaches to circumvent lack of power is frowned upon in the genomics community. The results are not secured and would need independent validation.

Minor:

1. Uncertain about the comment: "coding changes and host genotypes for HIV42 and HCV43, though these have much less variation than the pneumococcal population". It is difficult to consider a system that is more prone to diversity than RNA viruses, in particular HIV.

2. Use “pGWAS” and “hGWAS” to clarify where the GWAS is performed (pathogen/host).
3. The modeling of study power through Bayesian approaches is speculative and probably best to delete.

Reviewer #3 (Remarks to the Author):

The article by Lees et. al. is a much needed attempt to identify genetic factors of pneumococcal infection specifically bacterial meningitis both at the level of host and the pathogen.

The authors have used a variety of statistical and bioinformatics approaches to identify these candidate genes. Though the causality cannot be established from the results presented in the paper, it will certainly help in generating novel hypotheses for future experiments.

The paper does need some minor revisions that are listed below:

1. The CCDC33 gene is spelled correctly in the abstract but it needs correction in the rest of the paper, where it is incorrectly spelled as CDCC33.
2. The reference provided for CCDC33 gene is that of Gtex consortium for its expression in whole blood and the brain. When the gene was checked in the Gtex database, the results are that it is predominantly a testicular gene. This needs to be corrected, else needs an updated reference.
3. In the results section, reference is needed for the function of dacB gene.
4. If the results in Table 3 are from Gtex database, it needs to be mentioned.
5. Since there are so many abbreviations, it might be useful to provide a list at the beginning.

Reviewer #1 (Remarks to the Author):

This is a very interesting manuscript presenting a thorough analysis of genetic variation of both host and pathogen affecting meningococcal disease, with an impressive array of statistical tests and that finds several interesting genes, especially on the bacterial side. It is also very commendable to try to look for interactions, although unsurprising that it is currently underpowered.

The manuscript is however somewhat weaker in biological interpretation. It is written in many ways like a standard human genetics GWAS manuscript but this is not quite appropriate because the audience is different and because there are several interpretational issues that are specific to bacterial GWAS especially because of the key issue of population structure.

Thank you for the enthusiasm about the datasets and methodology in our manuscript, and especially for the imaginative suggestions below which we think have helped add to the interpretations possible from this data. We have also added background information and rewritten the text as necessary/suggested to increase the standard of our biological interpretation and readability by the target audience. In particular, we address the issue of population structure more directly in the results.

(1) it is important to "see" the bacterial data. This can be done by trees which show the distribution of carriage and disease strains.

This is an excellent point, and not including such a figure was an oversight on our part. We have added the suggested plot as a main figure (figure 3), as well as being available in an interactive form due to its large size and density of information (URL https://microreact.org/project/Spn_GWAS/9eb0bd5d).

(2) the heritability of bacterial genotypes seems very high and this should be put in a bit more context. Is an additive model appropriate and interpretable? It is also worth spelling out what would a heritability of 1.0 mean. how much heritability could be explained by the top PCs (which might in part be due to differences between the populations that carriage and disease were sampled from)? The authors assume that serotypes are likely to be important but presumably their importance is highly confounded with population structure, like for most other genes. Is there any evidence for effect, individually or collectively of serotype when this is controlled for? As it is quoting the figure of 0.45 seems a bit misleading and not particularly relevant to overall interpretation.

As you note in point 6), we agree that the human genetics and statistical genetics used could do with more explanation. We have therefore added more detail on the calculation and interpretation of these heritabilities, particularly noting the model used and the interpretation of its output. We believe the additive model is best for these purposes: genotypes have been explicitly coded as haploid (so there is no need to consider dominant/recessive models), and we have ignored any interaction terms as we feel that is beyond the scope of this work.

We have also used a recently updated release of the limix package to overhaul our analysis of bacterial heritability. This has allowed us to estimate the heritability of binary phenotypes using an appropriate error structure, which has lowered our heritability estimate. This also allowed us to perform heritability calculations with PC

covariates, serotype (as suggested here), and rare variants (as suggested in 5) within a single consistent framework.

Using this approach, we have added the suggested analyses to the first section of the results. Particularly, we take time to note the validity of the assumption that serotype is an important factor, and how we can test this. While many readers from the pneumococcal field would likely be looking out for a result in the opposite direction (i.e. that serotype does not fully explain invasive disease) we agree it is important to test whether we can find evidence for any independent effect. This section now also directly addresses the issue of population structure confounding.

(3) If the sampling has been done well, then the disease population should be thought of as a non-random subset of the carriage population (i.e. with specific genotypes amplified), rather than a separate population, since all disease isolates were carriage isolates before causing disease. Therefore difference in Tajima's D as well as the results on heritability should be discussed in this light. It seems from a brief reading that the controls are pretty good.

This is a good way to view this sampling, and certainly useful in the interpretation of both of these results. We tried to obtain the best matched control cohort we could, and by matching time, age (as well as possible) and country we feel that we were able to minimise sampling bias other than the phenotype in question. We have added paragraphs which specifically discuss the heritability and Tajima's D results with this interpretation of the sampling in mind. We have also noted this point early on in the introduction.

(4) It is notable that so many of the hits are for "missense burden", but this phrase is only ever used in the table, never in the methods and I would like to be more sure that I know exactly what was done here and also I would like this finding interpreted, both biologically and methodologically especially because of the very high p-values that have been obtained for rather obscure genes. One speculation is that because so much of the difference is confounded with phylogenetic (clonal) structure, the main GWAS signal that remains is rare mutations that happen repeatedly at the tips of the tree. Are the genes with the highest p-values particularly prone to missense variants?

We have clarified the methodology we used for this test, now explicitly noting the terms as they appear in table 1. We have added some general discussion of this finding in methodological and biological terms (thank you for your suggestions here). We have tested whether the hit genes were more prone to missense variants, but didn't find any evidence of this. We have added a paragraph at the end of the pGWAS section to discuss this finding methodologically and biologically.

We have also added more extensive speculation on the more obscure genes in table 1, using more previous literature and recent studies of coexpression in different pneumococcal growth conditions. We also note that in doing these further investigations we found that some of the genes previously reported in table 1 are commonly pseudogenes in the population (though were intact in the reference, hence their initial inclusion), so the missense mutations classification is unlikely to be meaningful. We have therefore removed these genes (and noted this in the methods).

One of the genes was actually a BOX repeat, which we have noted separately (as previously).

(5) Relating to point 4, The results on different number of lof and damaging mutations between disease and carriage isolates is really intriguing and could hold a key to understanding disease biology but this is never interpreted. It is even more so because some genes seem to go in the opposite direction (based on the missense table, if I am interpreting it correctly). It seems like there is a systematic difference in vigor between disease and carriage isolates, with disease isolates more vigorous. Presumably rare mutations are unlikely to be included in heritability, making the true genomic heritability higher than the estimate? What is the overall proportion of heritability that can be explained by aggregate LoF burden (with and without population structure control) and by individual genes. It would also be interesting to see the effect of LoF at different gene frequencies. Are very rare variants responsible for much of the the effect?

Thank you for these thoughts and suggestions. Along with the additions to point 4, we have addressed this question more fully by a number of new analyses, which are now in their own section of the results. As suggested, we have attempted to estimate the contribution to heritability from the burden of rare and very rare variation.

We used the updated heritability framework to perform the analyses you suggest in a manner consistent with our other results. We were not able to find any significant contribution through this technique. As we explain in the text, this is likely due to methodological rather than biological reasons.

We have added interpretation of our finding that burden rates vary by phenotype (which still holds when considering reviewer 2's point 1 below) into the results, to tie in with the extra discussion of the Tajima's D results added in response to point 4.

For individual genes, the remaining hit which is clearly affected by rare variation is *zmpD*. We have added a SFS plot stratified by phenotype for this gene specifically, and indeed find that very rare variants appear to be responsible for most of the effect.

(6) The human genetics section is at present rather hard going for people who are not familiar with the nitty gritty of today's human GWAS. It is hard to get a sense of how believable exciting the findings are, especially in terms of how surprising/convincing the eQTL and chromatin conformation capture analysis results are in terms of enriching/validating the signal. This section could perhaps be rewritten with a slightly broader audience in mind, e.g. highlighting entry points to the literature for comparable scans.

We had not appreciated the terseness of the language in this section, and the lack of introduction to more advanced techniques not familiar to a broader audience. We have added a clearer introduction to this section, and emphasised both a description of what eQTL and chromatin conformation capture analysis are, and how they add to the interpretation of our results. We have attempted to frame these additions in similar terms to the pGWAS presented immediately before, highlighting differences between this and hGWAS for clarity. We have also added in appropriate citations to the literature as suggested.

Reviewer #2 (Remarks to the Author):

There is an opportunity to jointly assess the infecting agent and the host in GWAS analyses. This is the stated goal in this work where the authors perform genetic association studies for the pathogen, for host susceptibility, and a genome-to-genome analysis. The target pathogen is pneumococcus, and the main phenotypes are meningitis and invasive disease. By aggregating data, the authors reach some thousand isolate genomes, some thousand host genotypes, and a ~400 samples that are doubly typed. Unfortunately, there is limited attention to the immense heterogeneity that is brought to the analysis – including the fact that some of the cohorts are from different populations, several organisms, multiple possible pneumococcal populations, and an unknown number of confounders that are not controlled for.

There are a number of hesitations in methodology: some of the analyses combine various data sources without careful management of data structure (starting with host ancestry and pathogen clonal structure surveys). Some of the analyses would have been more appropriate as meta-analyses than as combined cohorts.

In addition, lack of power pushes the work to slice and dice by nesting targeted, candidate analyses, within GWASs. This is generally not accepted by the genomic community except if there is strong downstream validation.

There is valuable data in this manuscript – but the current approach does not support the overall claims.

We would like to thank the reviewer for raising these concerns. We appreciate that the previous version of the manuscript paid insufficient attention to differences between the cohorts included, and how results may be affected by combining them. Overall, we have refocused the paper on our main cohort, and made it clear where additional cohorts have been added, and where they may differ in terms of phenotype/collection. We have added information about the distribution of phenotypes and genetic background by cohort. We have also added the meta-analysis approach suggested for the pGWAS.

It is appropriate to combine slightly different phenotypes which may still share some underlying mechanism. For example, considering bacteremia and meningitis – both infections involve invasion of the bloodstream (and in this case these diseases are nested, all meningitis cases will be bacteremia cases). So while combining cohorts may lose the ability to find genetics underlying meningitis specifically (for example, crossing the blood-brain barrier), it can raise the power to find genetics underlying invasive disease generally. In hGWAS, raising the sample size by combining related phenotypes has been shown to increase power to understand broader disease processes (doi:10.1038/ng.3926). So while combining cohorts with such differences does lose power to detect specific effects, the extra power for finding other effects makes it a worthwhile approach. In this manuscript we have collected and sequenced two novel bacterial cohorts, and two novel human cohorts. We believe the increased sample size, reduction of cohort specific effects and therefore the ability to try to replicate results in further cohorts at this stage (before validation in the lab) makes

this effort ,and analysis approach, worthwhile. We have made data from all of these individual cohorts and analyses available, as well as appropriate combined analysis (with the addition of meta-analysis, as suggested).

However, we agree that the previous version of the manuscript did not properly address this issue, and in particular that it may have been confusing exactly which samples and phenotypes were included in each analysis, and why. This has been clarified in this revision, and we keep our focus on our main Dutch cohort.

Regarding the management of data structure: we had included a plot of host ancestry as supplementary figure 25, and a discussion of how this was used for quality control is in the methods. We have now, as reviewer 1 also suggested, included an analysis of the clonal structure of the pathogen as a main figure. All the hGWAS was performed as a meta-analysis. As we also note below, the analyses in the pGWAS combining cohorts used cohort identity as a covariate (fixed effect), which is equivalent to combining results from cohorts in a meta-analysis. However, we have also now performed this analysis as a meta-analysis, and added results by cohort where appropriate (we note that tables S3-S5 show results for the single Dutch cohort).

We have controlled for confounders wherever we were able to, including host age, cohort (country), immunocompromised status of the host and genetic relatedness. Almost all studies of clinical data will have an unknown number of confounders and researchers must try their best to capture those most likely to be relevant and control for them. GWAS, given the agnostic approach and unidirectional cause, is likely less susceptible to confounding than comparable epidemiological studies (doi:10.1093/ije/dyg070; doi:10.1016/S0895-4356(00)00334-6). Given the amount of clinical metadata captured in our cohorts, we believe we have done an appropriate job of addressing this concern in the context of a case-control study.

Nested/candidate analyses were only used in the final genome-to-genome section, after first performing an agnostic scan where power was limited (which we provide an analysis and description of). We have made sure that we have clearly communicated the limitations in power inherent in this analysis, and than any results are speculative, needing further validation from future studies.

Meningitis is a severe disease, worthy of study, but its rarity does lead to challenges in collecting sufficient numbers of well-phenotyped and matched samples. Aggregating data is therefore necessary to achieve good type I and type II error rates. We believe the cohorts we have assembled and the analysis here is one of the best attempts at this important challenge, and will be helpful in generating candidates for further downstream validation.

We respond to the concerns also raised in your major points in detail below.

Specific comments:

1. Figure 2. I am surprised by the increase in most types of substitutions: including synonymous and intergenic. Could this be explained by more clonal nature of the invasive strains? It would be important to add a ratio analysis to the analysis of burden – where the LoF and damaging variants numbers are corrected by the synonymous (+intergenic) variation. From this analysis it is possible to surmise the excess burden. It would be useful to complement the Tajima's D test with a measure of selection such as dN/dS.

Regarding our observed increase of all types of mutation at the rare end of the site frequency spectrum in invasive isolates, oversampling of some rare clones in invasion would be one possible explanation for some of the effect in the lowest bin of the SFS histogram. While we also addressed this difference in our responses to reviewer #1's comments above, we have added the ratio analysis using synonymous mutations to correct the numbers per sample, as you suggest. We note that we did not include intergenic variation in this calculation, as some of it is likely to be under selection (doi:10.1534/genetics.116.195784); synonymous mutations are likely closest to neutral. With this correction, we still found evidence for a significant excess of burden for both LoF and damaging variants.

We also thank the reviewer for their suggestion of including a test of differing selection between carriage and disease cases. This is something we had considered testing originally, but upon reviewing literature of the applicability of dN/dS (or other tests of selection) within a bacterial population, there is evidence it would be 1) insensitive and difficult to interpret without a computationally intractable bootstrap (doi:10.1371/journal.pgen.1000304), 2) inappropriate without a temporally extensive sample set (doi:10.1016/j.jtbi.2005.08.037) 3) biased by recent polymorphisms (doi:10.1093/molbev/mst192). We also note that we are unable to accurately construct the ancestral state of mutations across the species due to extensive recombination. Considering Reviewer #1's point that disease samples are an expanded subset of carriage and therefore a single population is being analysed, these problems would make calculation and interpretation of a measure of selection, if not inappropriate, very challenging. We therefore feel the use of and interpretation of a measure of selection, particularly dN/dS, is likely to mislead readers. We think existing combination of analysis approaches are most appropriate, and have opted not to add a further complex analysis to an already wide-ranging manuscript.

2. I am worried about unaccounted for confounding when combining different cohorts and different locations (eg. Dutch meningitis plus South Africa invasive pneumococci)- a less worrisome approach would be as metaanalysis. At least, the behavior per cohort should be presented for the genes in Table 1.

Association through combining genotypes (our original approach), as long as ancestry is appropriately controlled for, and meta-analysis (the suggested approach) have been shown to give very similar results in hGWAS. Some differences between these approaches do exist.

As noted above, cohort differences are certainly an important concern, and one which we believe we have accounted for by using covariates for cohort, age and a combined kinship matrix between the populations. Potential disadvantages of using meta-analysis for combining pGWAS cohorts is that it is unclear what the effective sample size should be (this likely depends on the level of clonality, but has not yet

been explored), and treats populations as separate rather than adjusting for their shared ancestry. Also, recent evidence suggests that confounded GWAS summary statistics may lead to issues with a meta-analysis approach, and a combined analysis with genotypes may be preferable (doi:10.1101/532069).

However, as you make clear, it is certainly advantageous to be able to see the behaviour of signals in individual cohorts, which naturally comes out of a meta-analysis. We have therefore taken your suggestion and also performed a meta-analysis in the pGWAS. Our results were very similar, and as suggested we have added a table showing the per cohort behaviour of the results in table 1. The added figure 3 also helps visually display potential differences between the cohorts.

We had already taken this approach with the hGWAS, and present the single cohort results separately first.

3. Table 1. The use of multiple methods can be problematic, because it increases multiple testing. It should be made clear how p values will be adjusted both for the hypothesis tested, the number of variants and the number of techniques employed.

Thank you for this point, which we had not previously accounted for. We have now adjusted based on the number of techniques used, and removed results which did not pass this new threshold (as noted in table captions and the methods). While we would probably expect the values from different methods to be correlated, and a Bonferroni correction to be overly conservative, we applied this method as it is the most common and well-understood in the GWAS field.

4. Table 2, and elsewhere in the manuscript, there is analysis of “meningitis” – this is probably any type of meningitis, and not just pneumococcal. For the sake of consistency and clarity, the analysis of multi-cause meningitis should not be the object of analysis in this work. Severity should also be clarified – is it pneumococcal-proper, or all-cause? Such a heterogeneity could also result in spurious associations due to unaccounted for strata in the data.

Thank you for pointing to this potential confusion in phenotype. Indeed, one of the strengths of our cohort is using clinically confirmed meningitis, with the causal organism known. We have therefore ensured the focus is on pneumococcal meningitis by removing the all-cause meningitis results you note from tables 2 and 3 (now tables 3 and 4).

As we now note in the results, we did also attempt a severity analysis with pneumococcal meningitis only, but doing so was underpowered. The severity analysis presented in the table is therefore all-cause, as we now clarify in both tables. Our reasoning for combining severe meningitis cases from different organisms is that the enormous difference in host inflammatory response observed between these phenotypes may well be through the same host mechanism, irrespective of the specific infecting bacteria. One of the advantages of GWAS is that the phenotype can be anything meaningful, which from a clinical point of view severe meningitis clearly is. The hGWAS would therefore have the potential to tell us something about the host processes involved in increased inflammation in response to bacteria in the CSF.

To generate a spurious association with this number of samples would require both strong genetic association with invading bacteria, and a large difference in rates

of severe disease between different pathogens, neither of which appear to be the case. Additionally, studies in animals and patients have shown that unfavorable outcome results from the excessive host inflammatory response. So, the host inflammatory response is the common denominator.

We think it is unlikely that the association found is spurious due to unaccounted strata in the data, and we therefore argue that the analysis of severe meningitis caused by any organism is appropriate as long as the reasoning is well described, and the difference from other analyses clear. We have added text in the results to ensure this is the case.

5. There are several convenience analyses of other cohorts with various infectious disease causes and outcomes (Danish bacteremia, GenOSept and self-reported cases of sepsis in the UK biobank, 23andMe) that do not help with the message of the manuscript.

These results are most likely of interest to a more limited readership than the rest of the manuscript. We agree that these extra analyses probably added unnecessary and potentially confusing results to an already dense section. We have therefore removed these analyses where they appeared in the results.

One exception is the meta-analysis with self-reported meningitis cases in the UK biobank. We believe it is important to address how our results relate to the previous findings for bacterial meningitis (CFH and CA10), and this small analysis helps us relate these by matching phenotypes as well as possible, and then explain why there may be differences between studies. This helps keep cohort heterogeneity in the mind of the reader. In the rewritten section containing this result we have made it clear why we are doing this analysis, with respect to the main purpose of the manuscript.

6. The lack of power moved the authors to concentrate genome-genome analysis to few selected genes and also to “to reduce the multiple testing burden by reducing the dimension of the pathogen genotype”. In practice, these approaches to circumvent lack of power is frowned upon in the genomics community. The results are not secured and would need independent validation.

We agree that our results in the genome to genome analysis are necessarily limited in power and certainly would require independent validation. However we would also echo reviewer 3's point that these results may be useful for generating novel hypotheses for future experiments. We did first perform the ideal analysis of testing all possible interacting pairs, with an appropriately adjusted significance threshold. We have also clearly noted the limitations of power at the beginning of this section.

Given the prior evidence from many other studies about the importance of the chosen genes in disease, it is worthwhile testing these loci specifically, as long as we can effectively communicate this as a limitation of the sample size. Also, where large numbers of the tests are correlated, as is the case with the majority of the bacterial variants in this analysis, approaches which reduce this redundancy rather than making a simple p-value adjustment based on the assumption of independent tests are warranted. Looking at this from a biological perspective, testing whether host variation is associated with a particular strain of bacteria (rather than testing all the

thousands of correlated variants that define that strain individually) is a potentially useful approach.

However, we do agree that due to the lack of power we discuss that these results cannot be shown conclusively using this study alone. We have modified the wording of this section to ensure this is abundantly clear to the reader, and we have removed the part of the abstract that referenced these results outside of this necessary context. We believe these changes make our approach and its limitations transparent in the manuscript, while still potentially useful as a reference point for future studies.

Minor:

1. Uncertain about the comment: “coding changes and host genotypes for HIV42 and HCV43, though these have much less variation than the pneumococcal population”. It is difficult to consider a system that is more prone to diversity than RNA viruses, in particular HIV.

While the mutation rate in RNA viruses is higher than the pneumococcus, as noted by the reviewer, their shorter genomes lead to (in total) fewer observed variable sites in the alignment. So in the context of number of variant sites, there are far fewer than in our sampled population. We have now clarified this, rather than using the ambiguous term 'less variation'.

2. Use “pGWAS” and “hGWAS” to clarify where the GWAS is performed (pathogen/host).

We have made this clarification throughout.

3. The modeling of study power through Bayesian approaches is speculative and probably best to delete.

We have removed these sections from the results and methods as suggested.

Reviewer #3 (Remarks to the Author):

The article by Lees et. al. is a much needed attempt to identify genetic factors of pneumococcal infection specifically bacterial meningitis both at the level of host and the pathogen.

The authors have used a variety of statistical and bioinformatics approaches to identify these candidate genes. Though the causality cannot be established from the results presented in the paper, it will certainly help in generating novel hypotheses for future experiments.

We would agree with this summary of causality and generating hypotheses, and we are glad that you share our enthusiasm for the sharing of these datasets, methods and results. We have made all of the changes suggested below.

The paper does need some minor revisions that are listed below:

1. The CCDC33 gene is spelled correctly in the abstract but it needs correction in the rest of the paper, where it is incorrectly spelled as CDCC33.

Thanks for this correction – we have have replaced with the correct gene name throughout.

2. The reference provided for CCDC33 gene is that of Gtex consortium for its expression in whole blood and the brain. When the gene was checked in the Gtex database, the results are that it is predominantly a testicular gene. This needs to be corrected, else needs an updated reference.

We have corrected this sentence based on the results in the Gtex database.

3. In the results section, reference is needed for the function of dacB gene.

We have added the appropriate reference into the results.

4. If the results in Table 3 are from Gtex database, it needs to be mentioned.

We have added an explicit note of our use of the Gtex database, as well as the citation to the corresponding publication.

5. Since there are so many abbreviations, it might be useful to provide a list at the beginning.

We have added a glossary of terms to the introduction.

Reviewer #1 (Remarks to the Author):

This manuscript is definitely improved, I do like the new figure. :).

I still think a bit more context/logic could be spelled out for the human genetic data, to make it self contained for people who do not know about this stuff, in terms of the actual experiments done and the conclusions that can be drawn. For example, what are the conclusions (presumably tentative) that come from the observations below, viewed in the light of the GWAS results:

"the ME2 promoter variant

rs2850542 is an eQTL for the same gene in whole blood ($p = 5.9 \times 10^{-20}$ 492 , supplementary figure 9A) and specifically in monocytes

and from

. There was also evidence of chromatin

494 interaction of the variant location with SMAD4 again in a range of immune cell types

495 including monocytes, lymphocytes and neutrophils (supplementary figure 9B/C)

Reviewer #2 (Remarks to the Author):

The authors have made a good effort to revised the text. It is true that the analysis leads with a relatively rare disease and thus a real difficulty to gather sufficient numbers. However, there are still some doubts about the value of combining such extant cohorts (European and South African) – both in terms of handling human genetic diversity and association and in handling pneumococcal prevalent clones. Thus, I remain unconvinced that the signals are robust.

The authors also posit that meningitis and bacteremia are both processes that may share common susceptibility and pathogenesis. I disagree – many pneumococcal meningitis have a portal of entry locally: CSF leaks, suppurative otitis. Many pneumococcal bacteremia are of pulmonary origin. There is no strong argument for a common defect across these manifestations.

The statistics appeared pushed to their limits – hGWAS are reporting hits starting at 10^6 . It is also notable that rs116264669 in CCDC33 is retained in all three genetic studies but somehow not reported in Table 5. Regarding pGWAS, there are aspects in Table 2 and 3 that are confusing. In

Table 2, there are AF that are reported as “1” – it is unclear how genetic association can be done in that case – or otherwise what the AF is reporting. The meta analysis in Table 3 is reporting ORs of 1, and also ORs of different direction for some of the proposed associations.

Important in this studies, the current work has no detailed description of the participant cohorts and the key demographics, clinical and measurement data.

Reviewer #1 (Remarks to the Author):

This manuscript is definitely improved, I do like the new figure. :).

I still think a bit more context/logic could be spelled out for the human genetic data, to make it self contained for people who do not know about this stuff, in terms of the actual experiments done and the conclusions that can be drawn. For example, what are the conclusions (presumably tentative) that come from the observations below, viewed in the light of the GWAS results:

***"the ME2 promoter variant
rs2850542 is an eQTL for the same gene in whole blood ($p = 5.9 \times 10^{-20}$ 492 ,
supplementary figure 9A) and specifically in monocytes***

and from

. There was also evidence of chromatin

***494 interaction of the variant location with SMAD4 again in a range of immune cell
types***

495 including monocytes, lymphocytes and neutrophils (supplementary figure 9B/C)

As well as in the preamble to the hGWAS section describing these methods, we have also added specific information to the noted sections of the results on what precise experiment we performed, and the tentative conclusions that can be drawn from this extra information.

Reviewer #2 (Remarks to the Author):

The authors have made a good effort to revised the text. It is true that the analysis leads with a relatively rare disease and thus a real difficulty to gather sufficient numbers. However, there are still some doubts about the value of combining such extant cohorts (European and South African) – both in terms of handling human genetic diversity and association and in handling pneumococcal prevalent clones. Thus, I remain unconvinced that the signals are robust.

We believe that our work presents useful data, findings and methodology beyond the association signals. But, while we agree we have not been able to show these signals are robust using the presently available data, they will serve as useful hypothesis generation for future work. We have made further careful efforts to ensure the potential limitations of these findings are communicated to the readers throughout the manuscript.

The authors also posit that meningitis and bacteremia are both processes that may share common susceptibility and pathogenesis. I disagree – many pneumococcal meningitis have a portal of entry locally: CSF leaks, suppurative otitis. Many pneumococcal bacteremia are of pulmonary origin. There is no strong argument for a common defect across these manifestations.

The populations are indeed heterogeneous, and as such bacteremia and meningitis may not be the same phenotype. We do not claim that *all* meningitis and bacteremia cases share a common route (although most meningitis cases also have bacteremia – CSF leaks and suppurative otitis are very uncommon). However, both phenotypes usually involve invasion into the bloodstream, so we would be powered to find such mechanisms. While a study divided by exact phenotype would be ideal, the currently available numbers of cases are too low to permit this.

We do agree that we need to carefully acknowledge this difference in phenotype however, and have made additional efforts to clarify this, and its limitations, in the text. The previously suggested changes which include an analysis of these cohorts separately is still included.

The statistics appeared pushed to their limits – hGWAS are reporting hits starting at 10E6.

We have now noted in both the results and the caption of the hGWAS table (table 5) that these loci are only putatively associated, and noted the loci which pass the genome-wide significance threshold.

It is also notable that rs116264669 in CCDC33 is retained in all three genetic studies but somehow not reported in Table 5.

rs116264669 is positively associated in all studies, but did not reach significance in any individual study (hence, it did not meet the threshold to be included in table 5). This is likely due to its low MAF meaning that association in any individual cohort was underpowered.

Regarding pGWAS, there are aspects in Table 2 and 3 that are confusing. In Table 2, there are AF that are reported as “1” – it is unclear how genetic association can be done in that case – or otherwise what the AF is reporting.

We have clarified that AF in table 2 refers to the frequency of the gene in the population, not the individual variants.

The meta analysis in Table 3 is reporting ORs of 1, and also ORs of different direction for some of the proposed associations.

The reviewer is referring to the second and third rows of table 3.

In the second row (*pspC*), the OR of 1 and p-value of 0.8 in the Dutch cohort suggests this is a signal found only in the South African cohort. The presentation as a meta-analysis in this table (suggested by the reviewer in the previous revision) displays this information nicely, which is lost in table 2 alone. This is discussed in detail in the text.

In the third row (*spnTVRhsdS*), the differing OR directions again suggest differences between the two cohorts. The pooled analysis finds a different result, likely due to population structure effects in the linear mixed model used. Given that this is a specificity subunit of a restriction-modification system, and that these have previously been shown to vary independently of genetic background in *S. pneumoniae*, this interpretation is certainly

plausible in terms of the method used. However, overall this is a weak result, and therefore we do not draw any conclusions from it.

Important in this studies, the current work has no detailed description of the participant cohorts and the key demographics, clinical and measurement data.

These details were previously present in the supplementary methods, but we have now moved these to the methods section in the main text. We have also added further information when restricted to just the pneumococcal meningitis cases.